# Neural Flows: Efficient Alternative to Neural ODEs

**Marin Biloš**[1]*, **Johanna Sommer**[1], **Syama Sundar Rangapuram**[2],
**Tim Januschowski**[2], **Stephan Günnemann**[1]
[1]Technical University of Munich, [2]AWS AI Labs, Germany

## Abstract

Neural ordinary differential equations describe how values change in time. This is the reason why they gained importance in modeling sequential data, especially when the observations are made at irregular intervals. In this paper we propose an alternative by directly modeling the solution curves — the flow of an ODE — with a neural network. This immediately eliminates the need for expensive numerical solvers while still maintaining the modeling capability of neural ODEs. We propose several flow architectures suitable for different applications by establishing precise conditions on when a function defines a valid flow. Apart from computational efficiency, we also provide empirical evidence of favorable generalization performance via applications in time series modeling, forecasting, and density estimation.

## 1 Introduction

Ordinary differential equations (ODEs) are among the most important tools for modeling complex systems, both in natural and social sciences. They describe the *instantaneous change* in the system, which is often an easier way to model physical phenomena than specifying the whole system itself. For example, the change of the pendulum angle or the change in population can be naturally expressed in the differential form. Similarly, Chen et al. [11] introduce neural ODEs that describe how some quantity of interest represented as a vector $x$, changes with time: $\dot{x} = f(t, x(t))$, where $f$ is now a neural network. Starting at some initial value $x(t_0)$ we can find the result of this dynamic at any $t_1$:

$$x(t_1) = x(t_0) + \int_{t_0}^{t_1} f(t, x(t)) \, dt = \text{ODESolve}(x(t_0), f, t_0, t_1). \tag{1}$$

It is sufficient for $f$ to be continuous in $t$ and Lipschitz continuous in $x$ to have a unique solution, by the Picard–Lindelöf theorem [14]. This mild condition is already satisfied by a large family of neural networks. In most practically relevant scenarios, the integral in Equation 1 has to be solved numerically, requiring a trade-off between computation cost and numerical precision. Much of the follow up work to [11] focused on retaining expressive dynamics while requiring fewer solver evaluations [22, 37].

In the machine learning context we are given a set of initial conditions (often at $t_0 = 0$) and a loss function for the solution evaluated at time $t_1$. One example is modeling time series where the latent state is evolved in continuous time and is used to predict the observed measurements [16]. Here, unlike in physics for example, the function $f$ is completely unknown and needs to be learned from data. Thus, [11] used neural networks to model it, for their ability to capture complex dynamics. However, note that this comes at the cost of the ODE being non-interpretable.

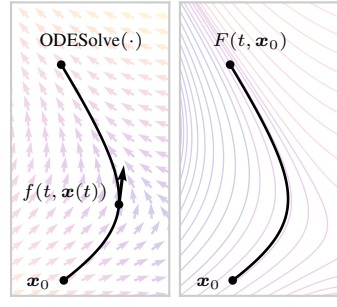

Figure 1: (Left) ODE requires numerical solver which evaluates $f$ at many points along the solution curve. (Right) Our approach returns the solutions directly.

---

*Work partially done during an internship at Amazon Research. Correspondence to: `bilos@in.tum.de`.

35th Conference on Neural Information Processing Systems (NeurIPS 2021).

Since solving an ODE is expensive, we want to find a way to keep the desired properties of neural ODEs at a much smaller computation cost. If we take a step back, we see that neural ODEs take initial values as inputs and return non-intersecting solution curves (Figure 1). In this paper we propose to model the solution curves directly, with a neural network, instead of specifying the derivative. That is, given an initial condition we return the solution with a single forward pass through our network. Straight away, this leads to improvements in computation performance because we avoid using ODE solvers altogether. We show how our method can be used as a faster alternative to ODEs in existing models [9, 16, 34, 69], while improving the modeling performance. In the following, we derive the conditions that our method needs to satisfy and propose different architectures that implement them.

## 2 Neural flows

In this section, we present our method, *neural flows*, that directly models the solution curve of an ODE with a neural network. For simplicity, let us briefly assume that the initial condition $x_0 = x(t_0)$ is specified at $t_0 = 0$. We handle the general case shortly. Then, Equation 1 can be written as $x(t) = F(t, x_0)$, where $F$ is the solution to the initial value problem, $\dot{x} = f(t, x(t)), x_0 = x(0)$. We will model $F$ with a neural network. For this, we first list the conditions that $F$ must satisfy so that it is a solution to some ODE. Let $F : [0, T] \times \mathbb{R}^d \to \mathbb{R}^d$ be a smooth function satisfying:

- i) $F(0, x_0) = x_0$, (initial condition)
- ii) $F(t, \cdot)$ is invertible, $\forall t$. (uniqueness of the solution given the initial value $x_0$; i.e., the curves specified by $F$ corresponding to different initial values should not intersect for any $t$)

There is an exact correspondence between a function $F$ with the above properties and an ODE defined with $f$ such that the derivative $\frac{d}{dt} F(t, x_0)$ matches $f(t, x(t))$ everywhere, given $x_0 = x(0)$ [47, Theorem 9.12]. In general, we can say that $f$ defines a vector field and $F$ defines a family of integral curves, also known as the *flow* in mathematics (not to be confused with normalizing flow). As $F$ will be parameterized with a neural network, condition i) requires that its parameters must depend on $t$ such that we have the identity map at $t = 0$.

Note that by providing $x_0$ we define a smooth trajectory $F(\cdot, x_0)$ — the solution to some ODE with the initial condition at $t_0 = 0$. If we relax the restriction $t_0 = 0$ and allow $x_0$ to be specified at an arbitrary $t_0 \in \mathbb{R}$, the solution can be obtained with a simple procedure. We first go back to the case $t = 0$ where we obtain the corresponding "initial" value $\hat{x}_0 := x(0) = F^{-1}(t_0, x_0)$. This then gives us the required solution $F(\cdot, \hat{x}_0)$ to the original initial value problem. Thus, we often prefer functions with an analytical inverse.

Finally, we tackle implementing $F$. The second property instructs us that the function $F(t, \cdot)$ is a diffeomorphism on $\mathbb{R}^d$. We can satisfy this by drawing inspiration from existing works on normalizing flows and invertible neural networks [e.g., 17, 2]. In our case, the parameters must be conditioned on time, with identity at $t = 0$. As a starting example, consider a linear ODE $f(t, x(t)) = Ax(t)$, with $x(0) = x_0$. Its solution can be expressed as $F(t, x_0) = \exp(At)x_0$, where $\exp$ is the matrix exponential. Here, the learnable parameters $A$ are simply multiplied by $t$ to ensure condition i); and given fixed $t$, the network behaves as an invertible linear transformation. In the following we propose other, more expressive functions suitable for applications such as time series modeling.

### 2.1 Proposed flow architectures

**ResNet flow.** A single residual layer $x_{t+1} = x_t + g(x_t)$ [30] bears a resemblance to Equation 1 and can be seen as a discretized version of a continuous transformation which inspired the development of neural ODEs. Although plain ResNets are not invertible, one could use spectral normalization [26] to enforce a small Lipschitz constant of the network, which guarantees invertibility [2, Theorem 1]. Thus, ResNets become a natural choice for modeling the solution curve $F$ resulting in the following extension — ResNet flow:

$$F(t, x) = x + \varphi(t)g(t, x), \qquad (2)$$

where $\varphi : \mathbb{R} \to \mathbb{R}^d$. This satisfies properties i) and ii) from above when $\varphi(0) = 0$ and $|\varphi(t)_i| < 1$; and $g : \mathbb{R}^{d+1} \to \mathbb{R}^d$ is an arbitrary contractive neural network ($\mathrm{Lip}(g) < 1$). One simple choice for $\varphi$ is a $\tanh$ function. The inverse of $F$ can be found via fixed point iteration similar to [2].

**GRU flow.** Time series data is traditionally modeled with recurrent neural networks, e.g., with a GRU [12], such that the hidden state $\boldsymbol{h}_{t-1}$ is updated at fixed intervals with the new observation $\boldsymbol{x}_t$:

$$\boldsymbol{h}_t = \text{GRUCell}(\boldsymbol{h}_{t-1}, \boldsymbol{x}_t) = \boldsymbol{z}_t \odot \boldsymbol{h}_{t-1} + (1 - \boldsymbol{z}_t) \odot \boldsymbol{c}_t, \tag{3}$$

where $\boldsymbol{z}_t$ and $\boldsymbol{c}_t$ are functions of the previous state $\boldsymbol{h}_{t-1}$ and the new input $\boldsymbol{x}_t$.

De Brouwer et al. [16] derived the continuous equivalent of this architecture called GRU-ODE (see Appendix A.1). Given the initial condition $\boldsymbol{h}_0 = \boldsymbol{h}(t_0)$, they evolve the hidden state $\boldsymbol{h}(t)$ with an ODE, until they observe new $\boldsymbol{x}_{t_1}$ at time $t_1$, when they use Equation 3 to update it:

$$\bar{\boldsymbol{h}}_{t_1} = \text{ODESolve}(\boldsymbol{h}_0, \text{GRU-ODE}, t_0, t_1), \quad \boldsymbol{h}_{t_1} = \text{GRUCell}(\bar{\boldsymbol{h}}_{t_1}, \boldsymbol{x}_{t_1}). \tag{4}$$

Here, we will derive the flow version of GRU-ODE. If we rewrite Equation 3 by regrouping terms: $\boldsymbol{h}_t = \boldsymbol{h}_{t-1} + (1 - \boldsymbol{z}_t) \odot (\boldsymbol{c}_t - \boldsymbol{h}_{t-1})$, we see that GRU update acts as a single ResNet layer.

**Definition 1.** *Let $f_z, f_r, f_c : \mathbb{R}^{d+1} \to \mathbb{R}^d$ be any arbitrary neural networks and let $z(t, \boldsymbol{h}) = \alpha \cdot \sigma(f_z(t, \boldsymbol{h})), r(t, \boldsymbol{h}) = \beta \cdot \sigma(f_r(t, \boldsymbol{h})), c(t, \boldsymbol{h}) = \tanh(f_c(t, r(t, \boldsymbol{h}) \odot \boldsymbol{h})),$ where $\alpha, \beta \in \mathbb{R}$ and $\sigma$ is a sigmoid function. Further, let $\varphi : \mathbb{R} \to \mathbb{R}^d$ be a continuous function with $\varphi(0) = \boldsymbol{0}$ and $|\varphi(t)_i| < 1$. Then the evolution of GRU state in continuous time is defined as:*

$$F(t, \boldsymbol{h}) = \boldsymbol{h} + \varphi(t)(1 - z(t, \boldsymbol{h})) \odot (c(t, \boldsymbol{h}) - \boldsymbol{h}). \tag{5}$$

**Theorem 1.** *A neural network defined by Equation 5 specifies a flow when the functions $f_z$, $f_r$ and $f_c$ are contractive maps, i.e., $\text{Lip}(f_.) < 1$, and $\alpha = \frac{2}{5}$, $\beta = \frac{4}{5}$.*

We prove Theorem 1 in Appendix A.3 by showing that the second summand on the right hand side in Equation 5 satisfies Lipschitz constraint making the whole network invertible. We also show that the GRU flow has the same desired properties as GRU-ODE, namely, bounding the hidden state in $(-1, 1)$ and having the Lipschitz constant of 2. Note that GRU flow (Equation 5) acts as a replacement to ODESolve in Equation 4. Alternatively, we can append $\boldsymbol{x}_t$ to the input of $f_z$, $f_r$ and $f_c$, which would give us a continuous-in-time version of GRU.

**Coupling flow.** The disadvantage of both ResNet flow and GRU flow is the missing analytical inverse. To this end, we propose a continuous-in-time version of an invertible transformation based on splitting the input dimensions into two disjoint sets $A$ and $B$, $A \cup B = \{1, 2, \ldots, d\}$ [17]. We copy the values indexed by $B$ and transform the rest conditioned on $\boldsymbol{x}_B$ which gives us the coupling flow:

$$F(t, \boldsymbol{x})_A = \boldsymbol{x}_A \exp(u(t, \boldsymbol{x}_B)\varphi_u(t)) + v(t, \boldsymbol{x}_B)\varphi_v(t), \tag{6}$$

where $u, v$ are arbitrary neural networks and $\varphi_u(0) = \varphi_v(0) = \boldsymbol{0}$. We can easily see that this satisfies condition i), and it is invertible by design regardless of $t$ [17]. Since some values stay constant in a single layer, we apply multiple consecutive transformations, choosing different partitions $A$ and $B$.

For all three models we can stack multiple layers $F = F_1 \circ \cdots \circ F_n$ and still define a proper flow since the composition of invertible functions is invertible, and consecutive identities give an identity.

We can think of $\varphi$ (including $\varphi_u, \varphi_v$) as a time embedding function that has to be zero at $t = 0$. Since it is a function of a single variable, we would like to keep the complexity low and avoid using general neural networks in favor of interpretable and expressive basis functions. A simple example is linear dependence on time $\varphi(t) = \boldsymbol{\alpha} t$, or $\tanh(\boldsymbol{\alpha} t)$ for ResNet flow. We use these in the experiments. An alternative, more powerful embedding consists of Fourier features $\varphi(t)_i = \sum_k \boldsymbol{\alpha}_{ik} \sin(\boldsymbol{\beta}_{ik} t)$.

## 2.2 On approximation capabilities

Previous works established that neural ODEs are $sup$-universal for diffeomorphic functions [76] and are $L^p$-universal for continuous maps when composed with terminal family [48]. A similar result also holds for affine coupling flows [75], whereas general residual networks can approximate any function [53]. The ResNet flow, as defined in Equation 2, can be viewed as an Euler discretization, meaning it is enough to stack appropriately many layers to uniformly approximate any ODE solution [48]. GRU flow can be viewed as a ResNet flow and coupling flow shares a similar structure, meaning that if we can set them to act as an Euler discretization we can match any ODE. However, this is of limited use in practice since we use finitely many layers, so the main focus of this paper is to provide the empirical evidence that we can outperform neural ODEs on relevant real-world tasks.

Other results [20, 81] consider limitations of neural ODEs in modeling general homeomorphisms (e.g., $x \mapsto -x$) and propose the solution that adds dimensions to the input $\boldsymbol{x}$. Such augmented

networks can model higher order dynamics. This can be explicitly defined through certain constraints for further improvements in performance and better interpretability [59]. We can apply the same trick to our models. However, instead of augmenting $\boldsymbol{x}$, a simpler solution is to relax the conditions on $F$ given the task. For example, if we do not need invertibility, we can remove the Lipschitz constraint in Equation 2. Since neural flows offer such flexibility, they might be of more practical relevance in these use cases.

## 3 Applications

In this section we review two main applications of neural ODEs: modeling irregularly-sampled time series and density estimation. We describe the existing modeling approaches and propose extensions using neural flows. In Section 4 we will use models presented here to qualitatively and quantitatively compare neural flows with neural ODEs.

### 3.1 Continuous-time latent variable models

Autoregressive [62, 70] and state space models [32, 68] have achieved considerable success modeling regularly-sampled time series. However, many real-world applications do not have a constant sampling rate and may contain missing values, e.g., in healthcare we have very sparse measurements at irregular time intervals. Here we describe how our neural flow models can be used in such scenario.

**Encoder.** In this setting, we are given a sequence of observations $\boldsymbol{X} = (\boldsymbol{x}_1, \ldots, \boldsymbol{x}_n)$, $\boldsymbol{x}_i \in \mathbb{R}^d$ at times $\boldsymbol{t} = (t_1, \ldots, t_n)$. To represent this type of data, previous RNN-based works relied on exponentially decaying hidden state [8], time gating [58], or simply adding time as an additional input [19]. More recently, various ODE-based models built on top of RNNs to evolve the hidden state between observations in continuous time, giving rise to, e.g., ODE-RNN [69], while outperforming previous approaches. Another model is GRU-ODE [16], which we already described in Equation 4. We proposed the GRU flow (Equation 5) that can be used as a straightforward replacement.

Lechner and Hasani [46] showed that simply evolving the hidden state with a neural ODE can cause vanishing or exploding gradients, a known issue in RNNs [3]. Thus, they propose using an LSTM-based [31] model instead. The difference to ODE-RNN [69] is using an LSTMCell and introducing another hidden state that is not updated continuously in time, which in turn allows gradient propagation via internal LSTM gating. To adapt this to our framework, we simply replace the ODESolve with the ResNet or coupling flow to obtain a neural flow model.

**Decoder.** Once we have a hidden state representation $\boldsymbol{h}_i$ of the irregularly-sampled sequence up to $\boldsymbol{x}_i$, we are interested in making future predictions. The ODE based models continue evolving the hidden state using a numerical solver to get the representation at time $t_{i+1}$, with $\boldsymbol{h}_{i+1} = \text{ODESolve}(\boldsymbol{h}_i, f, t_i, t_{i+1})$. With neural flows we can simply pass the next time point $t_{i+1}$ into $F$ and get the next hidden state directly. In the following we show how the presented encoder-decoder model is used in both the smoothing and filtering approaches for irregular time series modeling.

**Smoothing approach.** The given sequence of observations $(\boldsymbol{X}, \boldsymbol{t})$ is modeled with latent variables or states $(\boldsymbol{z}_1, \ldots, \boldsymbol{z}_n) \sim \mathbb{R}^h$, such that $\boldsymbol{x}_i \sim p(\boldsymbol{x}_i | \boldsymbol{z}_i)$, conditionally independent of other $\boldsymbol{x}_j$ [11, 69]. There is a predesignated prior state $\boldsymbol{z}_0$ at $t = 0$ from which the latent state is assumed to evolve continuously. More precisely, if $z_0$ is a sample from the initial latent state $\boldsymbol{z}_0$, then a latent state sample at any future time step $t$ is given by $z_t = F(t, z_0)$.

Since the exact inference on the initial state $\boldsymbol{z}_0$, $p(\boldsymbol{z}_0 | \boldsymbol{X}, \boldsymbol{t})$, is intractable, we proceed by doing approximate inference following the variational auto-encoder approach [11, 69]. We use an LSTM-based neural flow encoder that processes $(\boldsymbol{X}, \boldsymbol{t})$ and outputs the approximate posterior parameters $\boldsymbol{\mu}$ and $\boldsymbol{\sigma}$ from the last state, $q(\boldsymbol{z}_0 | \boldsymbol{X}, \boldsymbol{t}) = \mathcal{N}(\boldsymbol{\mu}, \boldsymbol{\sigma})$. The decoder returns all $z_i$ deterministically at times $\boldsymbol{t}$ with $F(t, z_0)$, with initial condition $z_0 \sim q(\boldsymbol{z}_0 | \boldsymbol{X}, \boldsymbol{t})$. For the latent state at an arbitrary $t_i$, the target is generated according to the model $\boldsymbol{x}_i \sim p(\boldsymbol{x}_i | \boldsymbol{z}_i)$. Given $p(\boldsymbol{z}_0) = \mathcal{N}(\boldsymbol{0}, \boldsymbol{1})$, the overall model is trained by maximizing the evidence lower bound:

$$\text{ELBO} = \mathbb{E}_{z_0 \sim q(\boldsymbol{z}_0 | \boldsymbol{X}, \boldsymbol{t}))}[\log p(\boldsymbol{X})] - \text{KL}[q(\boldsymbol{z}_0 | \boldsymbol{X}, \boldsymbol{t}) || p(\boldsymbol{z}_0)]. \tag{7}$$

Using continuous time models brings up multiple advantages, from handling irregular time points automatically to making predictions at any, and as many time points as required, allowing us to do

reconstruction, missing value imputation and forecasting. This holds whether we use neural flows or ODEs, but our approach is more computationally efficient, which matters as we scale to bigger data.

**Filtering approach.** In contrast to the previous approach, we can alternatively do the inference in an online fashion at each of the observed time points, i.e., estimating the posterior $p(z_i|x_{1:i}, t_{1:i})$ after seeing observations until the current time step $i$. This is known as filtering. Here, the prediction for future time steps is done by evolving the posterior corresponding to the final observed time point $p(z_n|X, t)$ instead of the initial time point $p(z_0|X, t)$, as was done in the smoothing approach.

In this paper, we follow the general approach suggested by De Brouwer et al. [16] for capturing non-linear dynamics. We use GRU flow (instead of GRU-ODE) for evolving the hidden state $h_i \in \mathbb{R}^h$ and we output the mean and variance of the approximate posterior $q(z_i|x_{1:i}, t_{1:i})$. The log-likelihood cannot be computed exactly under this model so [16] suggest using a custom objective that is the analogue to Bayesian filtering (see Appendix A.2 for details). Unlike [16], which needs to solve the ODE for every observation, our method only needs a single pass through the network per observation.

### 3.2 Temporal point processes

Sometimes temporal data is measured irregularly *and* the times at which we observe the events come from some underlying process modeled with temporal point processes (TPPs). For example, we can use TPPs to model the times of messages between users. One example type of behavior we want to capture is excitation [29], e.g., observing one message increases the chance of seeing other soon after.

A realization of a TPP on an interval $[0, T]$ is an increasing sequence of arrival times $t = (t_1, \dots, t_n)$, $t_i \in [0, T]$, where $n$ is a random variable. The model is defined with an intensity function $\lambda(t)$ that tells us how many events we expect to see in some bounded area [15]. The intensity has to be positive. We define the history $\mathcal{H}_{t_i}$ as the events that precede $t_i$, and further define the conditional intensity function $\lambda^*(t)$ which depends on history. For convenience, we can also work with inter-event times $\tau_i = t_i - t_{i-1}$, without losing generality. We train the model by maximizing the log-likelihood:

$$\log p(t) = \sum_i^n \log \lambda^*(t_i) - \int_0^T \lambda^*(s) \, ds. \tag{8}$$

Previous works [72] used autoregressive models (e.g., RNNs) to represent the history with a fixed-size vector $h_i$ [19]. The intensity function can correspond to a simple distribution [19] or a mixture of distributions [71]. Then the integral in Equation 8 can be computed exactly. Another possibility is modeling $\lambda(t)$ with an arbitrary neural network which requires Monte Carlo integration [6, 56]. On the other hand, Jia and Benson [34] propose a jump ODE model that evolves the hidden state $h(t)$ with an ODE and updates the state with new observations, similar to LSTM-ODE. In this case, obtaining the hidden state and solving the integral in Equation 8 can be done in a single solver call.

**Marked point processes.** Often, we are also interested in what type of an event happened at time point $t_i$. Thus, we can assign the observed type $x_i$, also called mark, and model the arrival times and marks jointly: $p(t, X) = p(t)p(X|t)$. Written like this, we can keep the model for arrival times as in Equation 8, and add a module that inputs the history $h_i$ and the next time point $t_{i+1}$ and outputs the probabilities for each mark type. The special case of $x_i \in \mathbb{R}^d$ is covered in the next section.

### 3.3 Time-dependent density estimation

Normalizing flows (NFs) define densities with invertible transformations of random variables. That is, given a random variable $z \sim q(z)$, $z \in \mathbb{R}^d$ and an invertible function $F : \mathbb{R}^d \to \mathbb{R}^d$, we can compute the probability density function of $x = F(z)$ with the change of variables formula [65]: $p(x) = q(z)|\det J_F(z)|^{-1}$, where $J_F$ is the Jacobian of $F$. As we can see, it is important to define a function $F$ that is easily invertible and has a tractable determinant of the Jacobian. One example is the coupling NF [17], which we used to construct the coupling flow in Equation 6. Other tractable models include autoregressive [41, 64] and matrix factorization based NFs [4, 40].

In contrast to this, Chen et al. [11] define the transformation with an ODE: $f(t, z(t)) = \frac{\partial}{\partial t} z(t)$. This allows them to define the instantaneous change in log-density as well as the continuous equivalent to the change of variables formula, giving rise to the continuous normalizing flow (CNF):

$$\frac{\partial}{\partial t} \log p(z(t)) = -\mathrm{tr}\left(\frac{\partial f}{\partial z(t)}\right), \quad \log p(x) = \log q(z(t_0)) - \int_{t_0}^{t_1} \mathrm{tr}\left(\frac{\partial f}{\partial z(t)}\right) dt, \tag{9}$$

where $t_0 = 0$ and $t_1 = 1$ are usually fixed. The neural network $f$ can be arbitrary as long as it gives unique ODE solutions. This offers an advantage when we need special structure of $f$ that cannot be easily implemented with the discrete NFs, e.g., in physics we often require equivariant transformations [5, 43]. Besides the cost of running the solver, calculating the trace at each step in Equation 9 becomes intractable as the dimension of data grows, so one resorts to stochastic estimation [27]. A similar approximation method is used for estimating the determinant in an invertible ResNet model [2]. We discuss the computation complexity in Appendix A.8. Again, if we consider a linear ODE, we can easily show that calculating the trace and calculating the determinant of the corresponding flow is equivalent (see Appendix A.7).

However, we are not interested in comparison between different normalizing flows for stationary densities [see e.g., 42], since *flow endpoints* $t_0$ and $t_1$ are always fixed; thus, our models would be reduced to the discrete NFs. Recently, Chen et al. [9] demonstrated how CNFs can evolve the densities in continuous time, with varying $t_0$ and $t_1$, which proves useful for spatio-temporal data. We will show how to do the same with our coupling flow, something that has not been explored before.

**Spatio-temporal processes.** We reuse the notation from Section 3.2 to denote the arrival times with $t$ and marks with $X$, $x_i \in \mathbb{R}^d$, which are now continuous variables. Values $x_i$ often correspond to locations of events, e.g., earthquakes [60] or disease outbreaks [57]. We use the temporal point processes from Section 3.2 to model $p(t)$, and are only left with the conditional density $p(X|t)$. Chen et al. [9] propose several models for this, the first one being the time-varying CNF where $p(x_i|t_i)$ is estimated by integrating Equation 9 from $t_0 = 0$ to observed $t_i$. Using our affine coupling flow as defined in Equation 6 we can write:

$$p(\boldsymbol{x}_i|t_i) = q(F^{-1}(t_i, \boldsymbol{x}_i))|\det J_{F^{-1}}(\boldsymbol{x}_i)|, \tag{10}$$

where $q$ is the base density (defined with any NF) and the determinant is the product of the diagonal values of the Jacobian w.r.t. $x_i$, which are simply $\exp$ terms from Equation 6 [17]. The density $p$ evolves with time, the same way as in the CNF model, but without using the solver or trace estimation. To generate new realizations at $t_i$, we first sample from $q$ to get $x_0 \sim q(x_0)$, then evaluate $F(t_i, x_0)$.

An alternative model, attentive CNF [9], is more expressive compared to the time-varying CNF and more efficient than jump ODE models [9, 34]. The probability density of $x_i$ depends on all the previous values $x_{j<i}$ through the attention mechanism [79]. In our model, we represent all the previous points $x_{j<i}$ with an attention encoder and define a conditional coupling NF $p(x_i|t_i, x_{j<i})$. We describe the full model in Appendix A.5. Both of the previous models can also use ResNet flow, but the benefits over ODEs vanish since the determinant and the inverse require iterative procedure.

# 4 Experiments

In this section we show that flow-based models can match or outperform ODEs at a smaller computation cost, both in latent variable time series modeling, as well as TPPs and time-dependent density estimation. To make fair comparison, we used recently introduced reparameterization trick for ODEs that allows faster mini-batching [9], and the semi-norm trick for faster backpropagation [38], making the models more competitive compared to the original works. In all experiments we split the data into train, validation and test set; train with early stopping and report results on test set. We use Adam optimizer [39]. For training we use two different machines, one with 3.4GHz processor and 32GB RAM and another with 61GB RAM and NVIDIA Tesla V100 GPU 16GB [52]. All datasets are publicly available, we include the download links and release the code that reproduces the results.[2]

**Synthetic data.** We compare the performance of neural ODEs and neural flows on periodic signals and data generated from autonomous ODEs. Full setup and results are presented in Appendix B. In short, we observe that training with adaptive solvers [18] is slower compared to fixed-step solvers, as expected. With the fixed step, however, we are not guaranteed invertibility [63], which can be an issue in, e.g., density estimation. Using the same setup, our models are an order of magnitude faster. Finally, neural ODEs struggle with non-smooth signals while neural flows perform much better, although they also only output smooth dynamics. Neural flows are also better at extrapolating, although none of the models excel in this task.

**Stiff ODEs.** The numerical approach to solving ODEs is not only slow but it can be unstable. This can happen when the ODE becomes stiff, i.e., the solver needs to take very small steps even though

---

[2]https://www.daml.in.tum.de/neural-flows

| | MuJoCo | Activity | | Physionet | |
|---|---|---|---|---|---|
| | MSE | MSE | Accuracy | MSE | AUC |
| Neural ODE | 8.403±0.142 | 6.390±0.136 | 0.756±0.013 | **4.833±0.078** | 0.777±0.012 |
| Coupling flow | **4.217±0.147** | 6.579±0.049 | 0.752±0.012 | 4.860±0.070 | **0.788±0.004** |
| ResNet flow | 5.147±0.171 | **6.279±0.098** | **0.760±0.004** | 4.903±0.125 | 0.784±0.010 |

Table 1: Test mean squared error (lower is better) and accuracy/area under curve (higher is better). Best result is bolded, result within one standard deviation is highlighted. Averaged over 5 runs.

the solution curve is smooth. For neural ODEs, it can happen that the target dynamic is known to be stiff or the latent dynamic becomes stiff during training.

To see the effects of this, we use the experiment from [24]. The ODE is given by: $\dot{x} = -1000x + 3000 - 2000e^{-t}$. We train a neural ODE model and a coupling flow to match the data, minimizing MSE. The data contains initial conditions and solutions, on small intervals with $t_2 - t_1 = 0.125$, $t \in [0, 15]$. The flow first finds the solution at $t_0 = 0$ and then solves for $t_2$ (Section 2). We evaluate on an extended time interval given $x_0 = 0$. Figure 2 shows that the neural

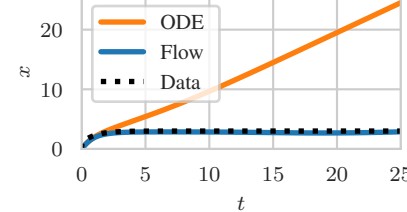

Figure 2: Flows handle stiffness better.

ODE with an adaptive solver does not match the correct solution, due to its stiffness. In contrast, flow captures the solution correctly, as expected, since it does not use a numerical solver.

**Smoothing approach.** Following [69], we use three datasets: Activity, Physionet, and MuJoCo. Activity contains 6554 time series of 3d positions of 4 sensors attached to an individual. The goal is to classify one of the 7 possible activities (e.g., walking, lying, etc.). Physionet [73] contains 8000 time series and 37 features of patients' measurements from the first 48 hours after being admitted to ICU. The goal is to predict the mortality. MuJoCo is created from a simple physics simulation "Hopper" [74] by randomly sampling initial positions and velocities and letting dynamics evolve deterministically in time. There are 10000 sequences, with 100 time steps and 14 features.

We use the encoder-decoder model (Section 3.1) and maximize Equation 7. We use the same number of hidden layers and the same size of latent states for both the neural ODE, coupling flow and ResNet flow, giving approximately the same number of trainable parameters. ODE models use either Euler or adaptive solvers and we report the best results. The results in Table 1 show the reconstruction error and the accuracy of prediction. For better readability, we scale MSE scores same as in [69]. Neural flows outperform ODE models everywhere (Physionet reconstruction within the confidence interval). We noticed that it is possible to further improve the results with bigger flow models but we focused on having similar sized models to show that we can get better results at a much smaller cost.

**Speed improvements.** In the smoothing experiment, our method offers more than two times speed-up during training compared to an ODE using an Euler method (Figure 3, different boxes corresponding to different datasets, grouped by experiment types). The gap is even larger for adaptive solvers. Note that Figure 3 shows an average time to run one training epoch which includes other operations, such as data fetching, state update etc. This shows that ODESolve contributes significantly to long training times. When comparing ODEs and flows alone, our method is much faster. In the following we will discuss the results from Figure 3 for other experiments as well as other results.

**Filtering approach.** Following De Brouwer et al. [16], we use clinical database MIMIC-III [35], pre-processed to contain 21250 patients' time series, with 96 features. We also process newly released MIMIC-IV [25, 36] to obtain 17874 patients. The details are in Appendix D.2. The goal is to predict the next three measurements in the 12 hour interval after the observation window of 36 hours.

Table 2 shows that our GRU flow model (Equation 5) mostly outperforms GRU-ODE [16]. Additionally, we show that the ordinary ResNet flow with 4 stacked transformations (Equation 2) performs worse. The reason might be because it is missing GRU flow properties, such as boundedness. Similarly, an ODE with a regular neural network does not outperform GRU-ODE [16]. Finally, we report that the model with GRU flow requires 60% less time to run one training epoch.

**Temporal point processes.** As we saw in Section 3.2, most of the TPP models consist of two parts: the encoder that processes the history, and the network that outputs the intensity. In the context of neural ODEs, we would like to answer: 1) whether having continuous state $\boldsymbol{h}(t)$ outperforms RNNs, and 2) if intertwining the hidden state evolution with the intensity outperforms other approaches. For this purpose we propose the following models based on continuous intensity and mixture distributions.

| | MIMIC-III | | MIMIC-IV | |
| --- | --- | --- | --- | --- |
| | MSE | NLL | MSE | NLL |
| GRU-ODE | 0.507±0.005 | **0.770±0.023** | 0.379±0.005 | 0.748±0.045 |
| ResNet flow | 0.508±0.007 | 0.779±0.023 | 0.379±0.005 | 0.774±0.059 |
| GRU flow | **0.499±0.004** | 0.781±0.041 | **0.364±0.008** | **0.734±0.054** |

Table 2: Forecasting on healthcare data averaged over 5 runs (lower is better).

| | | MOOC | | Reddit | | Wiki | |
| --- | --- | --- | --- | --- | --- | --- | --- |
| | Discrete GRU | -0.4448 | 2.7563 | -0.9299 | 1.8468 | -0.5832 | 8.0527 |
| Cont. | Jump ODE | 0.8710 | 4.6118 | 0.1308 | 3.6654 | -0.3115 | 10.6040 |
| | Coupling flow | 0.7694 | 5.5494 | -0.1263 | 3.6312 | -0.2807 | 9.7214 |
| | ResNet flow | **-1.2379** | 2.9466 | **-1.2962** | 2.3932 | -1.2907 | 10.4368 |
| Mix. | Jump ODE | -0.2626 | 3.0723 | -1.0907 | 1.9057 | **-1.3635** | **7.5537** |
| | Coupling flow | -0.4026 | **2.5877** | -1.0933 | **1.6817** | -1.2702 | 8.8018 |
| | ResNet flow | -0.5664 | 3.0005 | -1.0605 | 1.9491 | -1.1937 | 8.5489 |

Table 3: Test NLL for TPP (left columns, per dataset) and marked TPP (right columns); full results in Appendix C. Cont. denotes models with continuous intensity, and Mix. with mixture distribution.

Jump ODE evolves $\boldsymbol{h}(t)$ continuously together with the intensity function $\lambda(t) = g(\boldsymbol{h}(t))$ [34, 9], where $g$ is a neural network. The neural flow version replaces an ODE with our proposed flow models to evolve $\boldsymbol{h}(t)$ and uses Monte Carlo integration to evaluate Equation 8. Note that this operation can be parallelized unlike solving an ODE.

The mixture-based models keep the same continuous time encoders (ODEs and flows) but output the stationary log-normal mixture for the next arrival time. That is, instead of outputting the continuous intensity, they only use the hidden state at the last observation to define the probability density function [71]. As a baseline, we use a discrete GRU with the same mixture decoder.

We use both synthetic and real-world data, following [61, 71]. We generate 4 synthetic datasets corresponding to homogeneous, renewal and self-correcting processes. For real-world data, we collect timesteps of forum posts (Reddit), interactions of students with an online course system (MOOC), and Wiki page edits [44]. The details of the data are in Appendix D.3.

We report the test negative log-likelihood on real-world data in Table 3, for models trained both with and without marks. Full results, including synthetic data can be found in the Appendix C. We note that all the models capture the synthetic data, although continuous intensity models struggle compared to those with the mixture distribution. We can see this is the case for real-world data too, where the mixture distribution usually outperforms the corresponding continuous intensity model. In general, neural flows are better than ODE-based models, with the exception of one ODE model on Wiki dataset. We can conclude that having a continuous encoder is preferred to a discrete RNN because it can capture the irregular time sequence better. However, there is no benefit in parametrizing the intensity function in a continuous fashion, especially since this is a much slower approach.

Table 8 in Appendix C shows the comparison of wall clock times. Comparing only continuous intensity models we can see that Monte Carlo integration is faster than solving an ODE. As expected, using the mixture distribution gives the best performance. Thus, our flow models offer more than an order of magnitude faster processing compared to ODEs with continuous intensity. Figure 3 shows the difference for continuous models on the respective real-world datasets, the gap is even bigger if we include mixture-based models, where the speed-up is over an order of magnitude.

**Spatial data.** We compare the continuous normalizing flows with our continuous-time version of the coupling NF on time-dependent density estimation. We use two versions of each model: time-varying and attentive, as described in Section 3.3. Following Chen et al. [9], we use locations of bike rentals (Bikes), Covid cases for the state of New Jersey [77], and earthquake events in Japan (EQ) [78].

Results in Table 4 show the test NLL for spatial data, that is, we do not report the TPP loss since this is shared between models. Our continuous coupling NF models perform better on all datasets. Since affine coupling is a simple transformation, we require bigger models with more parameters. At the same time, our models are still more than an order of magnitude faster. Adapting some other, more expressive normalizing flows to satisfy flow constraints might reduce the number of parameters.

| | Bikes | Covid | EQ |
|---|---|---|---|
| Time-var. CNF | 2.315 | 1.984 | 1.709 |
| Attentive CNF | 2.371 | 1.973 | 1.668 |
| Time-var. coupling | **2.280** | **1.916** | 1.633 |
| Attentive coupling | 2.330 | 1.926 | **1.457** |

Table 4: Test NLL for spatial datasets.

Figure 3: Comparing per-epoch wall-clock times. Each box is dataset (order by appearance in text).

## 5 Discussion

In this paper we presented neural flows as an efficient alternative to neural ODEs. We retain all the desirable properties of neural ODEs, without using numerical solvers. Our method outperforms the ODE based models in time series modeling and density estimation, at a much smaller computation cost. This brings the possibility to scale to larger datasets and models in the future.

**Other related work.** Early works on approximating the ODE solutions without numerical solvers used splines or radial basis functions [55, 50], or functions similar to modern ResNets [45]. More recently, [66] approximate the solution by minimizing the error of the solution points and of the boundary condition. Unlike these approaches, we do not approximate the solution to some given ODE but learn the solutions which corresponds to learning the unknown ODE. Also, our method guarantees that we always define a proper flow, as is required in certain applications.

A similar problem is modeling the solutions to partial differential equations, e.g., with a model that is analogous to the classical discrete encoder-decoder [49]. Although we cannot compare these two settings directly, one could use our method to enhance modeling PDE solutions.

ResNets were initially recognized as a discretization of dynamical systems [51, 80] and were used to tackle infinite depth [1, 54], stability [13, 28] and invertibility [7, 33]. We take a different approach and propose modified ResNets, among other, avoiding any iterative procedure. ResNets also lead to neural ODEs which have memory efficient backpropagation as one of the main features [21, 11]. Further, to combat solver inefficiency, many improvements have been proposed, such as adding regularization [22, 24, 37], improving training [23, 38, 82] and having faster inference [67].

**Limitations.** Defining a flow automatically defines an ODE, but since many ODEs do not have closed-form solutions, we cannot always find the *exact* flow corresponding to a particular ODE. This is usually not an issue since in most applications, such as those presented in Section 3, it is sufficient for both neural ODEs and neural flows to approximate an unknown dynamic. However, if we restrict ourselves to autonomous ODEs (fixed vector field in time), we cannot define a general neural flow that satisfies this condition. We further discuss this in Appendix A.6 and present a potential solution that involves a simple regularization.

Since neural ODEs reuse the same function $f$ in the solver, essentially defining *implicit layers*, they can be more parameter efficient. Sometimes we might need more parameters to represent the same dynamic, as we observed in the density estimation task. But even here, the results show neural flows are more efficient. In the special setting with limited memory, we can resort to existing solutions [10].

**Future work.** In this work we designed neural flow models as invertible functions that satisfy initial condition using simple dependence on time. Although these models already outperform neural ODEs, it would be interesting to see if there are other ways to define a neural flow, and whether these architectures can outperform the ones we proposed here.

We applied our method to the main applications of neural ODEs: time series modeling and density estimation. In the future we hope to see neural flows adapted for other use cases as well. Investigating flows that define the higher order dynamics might also be of interest.

**Broader impact.** We introduced a new method to replace neural ODEs. As such, it has a wide variety of potential applications, some of which we explored in this paper. We used several healthcare datasets and hope to see further applications of our method in this domain. At the same time, it is important to pay attention to data privacy and fairness when building such models, especially for sensitive applications, such as healthcare. One of the main benefits of our method is the reduced computation cost, which may imply energy savings.

## Acknowledgments

We would like to thank Oleksandr Shchur for helpful discussions.

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
