# A Theoretical background

## A.1 GRU-ODE definition

De Brouwer et al. [16] define the continuous time GRU-ODE model as an ODE that is solved for hidden state $\boldsymbol{h}(t)$:

$$\frac{\mathrm{d}\boldsymbol{h}(t)}{\mathrm{d}t} = (1 - \boldsymbol{z}(t)) \odot (\boldsymbol{c}(t) - \boldsymbol{h}(t)). \tag{11}$$

With new observation $\boldsymbol{x}$, the hidden state is updated with discrete GRU (Equation 3), and between two observations we solve the ODE given by Equation 11.

The interesting properties of this model are:

    i) Boundedness: hidden state $\boldsymbol{h}(t)$ stays within range $(-1, 1)$,

    ii) Continuity: GRU-ODE is Lipschitz continuous with Lipschitz constant 2.

In Appendix A.3 we show how our GRU flow model has the same properties without the need to use numerical solvers.

## A.2 Training loss for GRU-ODE-Bayes

De Brouwer et al. [16] define an objective that mimics the Bayesian filtering. It consists of two parts:

$$\mathcal{L} = \mathcal{L}_{\mathrm{pre}} + \lambda \mathcal{L}_{\mathrm{post}}, \tag{12}$$

where $\mathcal{L}_{\mathrm{pre}}$ is masked negative log-likelihood and $\mathcal{L}_{\mathrm{post}}$ is the Bayesian part of the loss. The model outputs the normal distribution for the observations, conditional on hidden state $\boldsymbol{h}(t)$. Since only some features are observed at a time, we mask out the missing values when calculating $\mathcal{L}_{\mathrm{pre}}$. We denote our predicted distribution with $p_{\mathrm{pre}}$, and predicted distribution after updating the state with $p_{\mathrm{post}}$. Now, the Bayesian update can be written as $p_{\mathrm{Bayes}} \propto p_{\mathrm{pre}} \cdot p_{\mathrm{obs}}$, with $p_{\mathrm{obs}}$ being the noise of the observations. $\mathcal{L}_{\mathrm{post}}$ is defined as a KL-divergence between $p_{\mathrm{Bayes}}$ and $p_{\mathrm{post}}$. This can be calculated in closed-form for normal distribution.

## A.3 Proof of Theorem 1

**Preliminaries.** Function $f$ has the Lipschitz constant $L$ if $|f(x) - f(y)| \le L|x - y|, \forall x, y$. We first derive a few useful inequalities.

For the sum of two Lipschitz functions $f + g$, the following holds:

$$\begin{aligned}
|f(x) + g(x) - f(y) - g(y)| &\le |f(x) - f(y)| + |g(x) - g(y)| \\
&\le \mathrm{Lip}(f)|x - y| + \mathrm{Lip}(g)|x - y| \\
&\le (\mathrm{Lip}(f) + \mathrm{Lip}(g))|x - y|,
\end{aligned} \tag{13}$$

by the triangle inequality and the definition of the Lipschitz function. Similarly, for the product of two Lipschitz functions $f \cdot g$, the following holds:

$$\begin{aligned}
|f(x)g(x) - f(y)g(y)| &= |f(x)g(x) + f(x)g(y) - f(x)g(y) - f(y)g(y)| \\
&= |f(x)(g(x) - g(y)) + g(y)(f(x) - f(y))| \\
&\le |f(x)||g(x) - g(y)| + |g(y)||f(x) - f(y)| \\
&\le |f(x)| \cdot \mathrm{Lip}(g) \cdot |x - y| + |g(y)| \cdot \mathrm{Lip}(f) \cdot |x - y|. \\
&= (|f(x)| \cdot \mathrm{Lip}(g) + |g(y)| \cdot \mathrm{Lip}(f))|x - y|.
\end{aligned} \tag{14}$$

If $f$ and $g$ are bounded, we can bound the above term too.

Let $f$ be contractive function, $\mathrm{Lip}(f) < 1$. Then, for the composition of functions $\sigma \circ f$, where $\sigma(x) = (1 + \exp(-x))^{-1}$ is the sigmoid activation, the following holds:

$$|\sigma(f(x)) - \sigma(f(y))| \le \mathrm{Lip}(\sigma)|f(x) - f(y)| = \frac{1}{4}|f(x) - f(y)| \le \frac{1}{4}|x - y|,$$

where we used $\mathrm{Lip}(\sigma) = \max(\sigma') = \frac{1}{4}$, by the mean value theorem. Similarly, $\mathrm{Lip}(\tanh) = 1$.

*Proof.* (**Theorem 1**)

Equation 3 defines GRU as: $z_t \odot h_{t-1} + (1 - z_t) \odot c_t$. Since $z_t$ is defined as $\sigma(f_c(\cdot))$, and acts as a gate, we can equivalently define GRU with: $(1 - z_t) \odot h_{t-1} + z_t \odot c_t$. This will slightly simplify further calculations. Then, the GRU flow is defined as:

$$F(t, \boldsymbol{h}) = \boldsymbol{h} + \varphi(t) \odot z(t, \boldsymbol{h}) \odot (c(t, \boldsymbol{h}) - \boldsymbol{h}). \tag{5}$$

$F$ is invertible when the second summand on the right hand side is a contractive map, i.e., has a Lipschitz constant smaller than one. Since $\varphi(t)$ is bounded to $[0, 1]$ and does not depend on $\boldsymbol{h}$, we only need to show that $z(t, \boldsymbol{h}) \odot (c(t, \boldsymbol{h}) - \boldsymbol{h})$ is contractive. From here, we denote with $x$ and $y$ the input to our functions.

Following Definition 1, let $r(x) = \beta \cdot \sigma(f_r(x))$, with $\mathrm{Lip}(f_r) < 1$. Then we can write:

$$\begin{aligned}
|r(x) - r(y)| &= |\beta \cdot \sigma(f_r(x)) - \beta \cdot \sigma(f_r(y))| \\
&\leq \beta|\sigma(f_r(x)) - \sigma(f_r(y))| \\
&\leq \frac{1}{4}\beta|f_r(x) - f_r(y)| \\
&< \frac{1}{4}\beta|x - y|.
\end{aligned} \tag{15}$$

Similarly, for $z(x)$, where $z(x) = \alpha \cdot \sigma(f_z(x))$, and $\mathrm{Lip}(f_z) < 1$:

$$|z(x) - z(y)| \leq |\alpha \cdot \sigma(f_z(x)) - \alpha \cdot \sigma(f_z(y))| < \frac{1}{4}\alpha|x - y|. \tag{16}$$

Then for $c(x) = \tanh(f_c(r(x) \cdot x))$, with $\mathrm{Lip}(f_c) < 1$, we can write:

$$\begin{aligned}
|c(x) - c(y)| &= |\tanh(f_c(r(x) \cdot x)) - \tanh(f_c(r(y) \cdot y))| \\
&\leq |f_c(r(x) \cdot x) - f_c(r(y) \cdot y)| \\
&< |r(x) \cdot x - r(y) \cdot y| \\
&< (\underbrace{|r(x)|}_{<\beta} \cdot \underbrace{\mathrm{Lip}(\mathrm{Id})}_{=1} + \underbrace{|x|}_{<1} \cdot \underbrace{\mathrm{Lip}(r)}_{<\frac{1}{4}\beta})|x - y|,
\end{aligned} \tag{17}$$

where we used Equation 14 in the last line. Then $\mathrm{Lip}(c) < \frac{5}{4}\beta$. Now, for $c(x) - x$, and using Equation 13, we write:

$$|c(x) - x - c(y) + y| \leq (\mathrm{Lip}(c) + 1)|x - y|, \tag{18}$$

meaning the whole term has Lipschitz constant $\frac{5}{4}\beta + 1$. Finally, for the term on the right hand side of Equation 5, the following holds:

$$\begin{aligned}
&|z(x)(c(x) - x) - z(y)(c(y) - y)| \\
&< (\underbrace{|z(x)|}_{<\alpha} \cdot \underbrace{\mathrm{Lip}(c(x) - x)}_{<\frac{5}{4}\beta+1} + \underbrace{|c(x) - x|}_{<2} \cdot \underbrace{\mathrm{Lip}(z(x))}_{<\frac{1}{4}\alpha})|x - y|.
\end{aligned}$$

If we set $\alpha = \frac{2}{5}$, $\beta = \frac{4}{5}$, then the Lipschitz constant is smaller than 1, as required. $\qquad\square$

### A.3.1 Properties of GRU flow

Our GRU flow has the same desired properties as GRU-ODE:

    i) Boundedness: hidden state $\boldsymbol{h}$ stays within range $(-1, 1)$,

    ii) Continuity: the whole transformation $\boldsymbol{h} + g(\boldsymbol{h})$ has Lipschitz constant $1 + \mathrm{Lip}(g) \leq 2$.

The gating mechanism in discrete GRU helps with gradient propagation to enable learning long-term dependencies. We emphasize that both GRU flow and GRU-ODE update the hidden state in two distinct ways: 1) with discrete GRU when the new observation arrives, and 2) with continuous GRU between observations. Thus, the gates $z$ and $r$ do not have the same interpretation in discrete GRUCell and in continuous GRU flow or GRU-ODE.

The same way, scalars $\alpha$ and $\beta$ should not be interpreted as bounds to how much information can pass, but as a way to ensure invertibility. GRU flow has the ability to keep the old state $\boldsymbol{h}$, and does so at the initial condition $t = 0$, but can also overwrite it completely.

### A.4 ODE reparameterization

The ODESolve operation is usually implemented such that it takes scalar start and end times, $t_0$ and $t_1$. However, we are often interested in processing the data in batches, to get speed-up from parallelism on modern hardware. When the previous works [11, 69, 16] received the vectors of start and end times, e.g., $\boldsymbol{t}_0 = [0, 0, 0]$ and $\boldsymbol{t}_1 = [5, 1, 4]$, they would concatenate all the values into a single vector and sort them to get a sequence of strictly ascending times, e.g., $[0, 1, 4, 5]$. The solver would then first solve $0 \to 1$, then $1 \to 4$, and finally $4 \to 5$. Note that for the element in the batch with the largest end time, this requires calling ODESolve multiple times (number of unique time values), instead of only once. Without this procedure, the adaptive solver could take larger steps then the ones imposed by the current batch, meaning we would get better performance.

Chen et al. [9] propose a reparameterization, such that, instead of solving the ODE on the interval $t \in [0, t_{\max}]$, they solve it on $s \in [0, 1]$, with $s = t/t_{\max}$. For the batch of size $n$, the joint system is:

$$\frac{\mathrm{d}}{\mathrm{d}s} \begin{bmatrix} \boldsymbol{x}_1 \\ \boldsymbol{x}_2 \\ \vdots \\ \boldsymbol{x}_n \end{bmatrix} = \begin{bmatrix} t_1 f(st_1, \boldsymbol{x}_1) \\ t_2 f(st_2, \boldsymbol{x}_2) \\ \vdots \\ t_n f(st_n, \boldsymbol{x}_n) \end{bmatrix}.$$

This allows solving the system in parallel, in contrast to previous works. We used this reparameterization in all of our experiments.

### A.5 Attentive normalizing flow

We follow the setup from Section 3.3, denoting times with $\boldsymbol{t} = (t_1, \ldots, t_n)$, and marks with $\boldsymbol{X} = (\boldsymbol{x}_1, \ldots, \boldsymbol{x}_n)$, $\boldsymbol{x}_i \in \mathbb{R}^d$. We define the self-attention layer, following [79], as:

$$\text{SelfAttention}(\boldsymbol{X}) = \text{Attention}(\boldsymbol{Q}, \boldsymbol{K}, \boldsymbol{V}) = \text{softmax}\left(\frac{\boldsymbol{Q}\boldsymbol{K}^T}{\sqrt{d_k}}\right)\boldsymbol{V}, \tag{19}$$

where $\boldsymbol{Q} \in \mathbb{R}^{n \times d_k}, \boldsymbol{K} \in \mathbb{R}^{n \times d_k}, \boldsymbol{V} \in \mathbb{R}^{n \times d_v}$ are matrices that we obtain by transforming each element $\boldsymbol{x}_i$ of $\boldsymbol{X}$ by a neural network. Chen et al. [9], in their attentive CNF model, define the function $f$ from Equation 9 for each $\boldsymbol{x}_i$, as the $i$th output of $\text{Attention}$ function. It is important that elements $\boldsymbol{x}_j$, $j > i$, do not influence $\boldsymbol{x}_i$ to ensure we have a proper temporal model. This is achieved by placing $-\infty$ for values above the diagonal of the $\boldsymbol{Q}\boldsymbol{K}^T$ matrix so that $\text{softmax}$ returns zero on these places.

Discrete normalizing flows cannot define the transformation using attention and have tractable determinant of the Jacobian at the same time. However, since we actually need an autoregressive model, i.e., the dependence is strictly on the past values, not future, we can define a model similar to attentive CNF. We use Equation 19 with diagonal masking to embed the history of all the elements that preceded $\boldsymbol{x}_i$: $\boldsymbol{h}_i = \text{SelfAttention}(\boldsymbol{X}_{1:i-1})$. This is in contrast to [9], who used $\boldsymbol{X}_{1:i}$. Then, the conditioning vector $\boldsymbol{h}_i$ is used as an additional input to neural networks $u$ and $v$ from Equation 6, essentially defining a conditional affine coupling normalizing flow.

### A.6 Autonomous ODEs

Autonomous differential equations are defined with a vector field that is fixed in time $\dot{\boldsymbol{x}} = f(\boldsymbol{x}(t))$. Note that function $f$ does not depend on time $t$ like before. Therefore, the conditions i) and ii) from Section 2 are not enough to define the corresponding flow. To be precise, the flow $F$ defines an autonomous ODE if it satisfies the additional condition:

iii) $F(t_1 + t_2, \boldsymbol{x}_0) = F(t_2, F(t_1, \boldsymbol{x}_0))$,

meaning that solving for $t_1$ first, then $t_2$, is the same as solving for $t_1 + t_2$, given initial condition $\boldsymbol{x}_0$.

More formally, we defined flow $F$ on set $\mathbb{R}^d$ as a group action of the additive group $G = (\mathbb{R}, +)$ (elements being time points). Equivalently, group action of $G$ on $\mathbb{R}^d$ is a group homeomorphism from $G$ to $\text{Sym}(\mathbb{R}^d)$ (symmetric group, bijective functions and composition $(\phi, \circ)$), i.e., some function $\varphi : G \to \text{Sym}(\mathbb{R}^d)$ maps time $t$ to parameters of an invertible neural network $\phi$, with

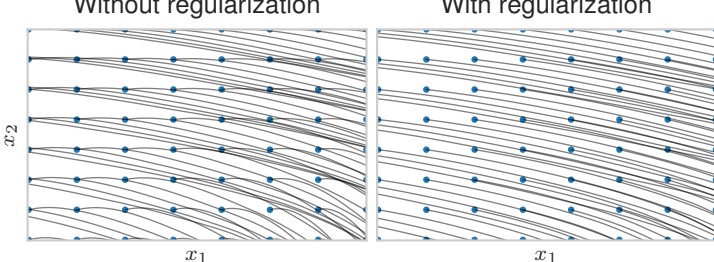

Figure 4: Comparison of the phase space for the same model trained with and without the autonomous regularization (Equation 20). Dots denote initial conditions. Note that the overlapping dynamic does not mean the solutions are not unique, only that the vector field is dependent on time.

$\varphi(t_1 + t_2) = \varphi(t_1) \circ \varphi(t_2)$. Identity element of $G$, $0$ is mapped to an identity function, inverse $-t$ is mapped to an inverse function.

It's clear that our proposed architectures from Section 2 do not satisfy condition iii), unless we redefine it to allow time-dependence. Therefore, one way to satisfy iii) is to have $\frac{d}{dt}F$ independent of time. Note, however, that if we define the ResNet flow as $\boldsymbol{x}_t := F(t, \boldsymbol{x}_0) = \boldsymbol{x}_0 + t \cdot h(\boldsymbol{x}_0)$, then even though time disappears from the derivative $\frac{d}{dt}F$, the derivative is expressed in terms of $\boldsymbol{x}_0$, not $\boldsymbol{x}_t$. This means time is still implicitly included since starting at different $\boldsymbol{x}_0$ gives different values.

Matrix exponential $\exp(\boldsymbol{A}t)\boldsymbol{x}$, as a solution to a linear ODE: $\dot{\boldsymbol{x}} = \boldsymbol{A}\boldsymbol{x}$, is one example of a closed-form solution to an autonomous ODE. Another potential *autonomous flow* is of the form $\boldsymbol{x} + \varphi(t)$, but not $g(\boldsymbol{x}) + \varphi(t)$, since this does not satisfy initial condition or $g$ must depend on time. To the best of our knowledge, there is no general neural flow parametrization that can capture all autonomous ODEs. Therefore, we can try to learn the desired behavior instead of guaranteeing it.

We can add the penalty to our loss that directly corresponds to condition iii). Given the loss function $\mathcal{L}$ and the current batch of $n$ elements $\boldsymbol{X} \in \mathbb{R}^{n \times d}$, $\boldsymbol{t} \in \mathbb{R}^n$, where we can represent each $t_i \in \boldsymbol{t}$ as $t_i = t_i^{(1)} + t_i^{(2)}$, with $t_i^{(1)}, t_i^{(2)}$ uniformly sampled on $[0, t_i]$, the total loss is:

$$\mathcal{L}_{\text{total}} = \mathcal{L} + \gamma \frac{1}{n} \sum_i (F(t_i, \boldsymbol{x}_i) - F(t_i^{(2)}, F(t_i^{(1)}, \boldsymbol{x}_i)))^2, \qquad (20)$$

where $\gamma$ is some positive value. The second term penalizes flows that do not satisfy iii), meaning we should get the flow that is closer to the underlying autonomous ODE. This can be calculated in parallel to other computations.

Figure 4 shows the comparison between learning the data generated from an autonomous ODE (see next section for data details), using the regularization as defined in Equation 20 and without such regularization. We can see that the base model already learns good behavior but when we include the regularization, the trajectories overlap less frequently.

## A.7 Linear ODE and change of variables

Consider a linear ODE $f(t, \boldsymbol{z}(t)) = \boldsymbol{A}\boldsymbol{z}(t)$, with $\boldsymbol{z}(0) = \boldsymbol{z}$ and $\boldsymbol{z}(1) = \boldsymbol{x}$. Solving the ODE $0 \to 1$ is the same as calculating $\exp(\boldsymbol{A})\boldsymbol{z}$, where $\exp$ is the matrix exponential. Suppose that $\boldsymbol{z} \sim q(\boldsymbol{z})$, then the distribution $p(\boldsymbol{x})$ that we get by transforming $\boldsymbol{x}$ with an ODE is defined as:

$$\log p(\boldsymbol{x}) = \log q(\boldsymbol{z}) - \int_0^1 \text{tr}\left(\frac{\partial f}{\partial \boldsymbol{z}(t)}\right) dt = \log q(\boldsymbol{z}) - \text{tr}(\boldsymbol{A}), \qquad (21)$$

or simply: $p(\boldsymbol{x}) = q(\boldsymbol{z}) \exp(\text{tr}(\boldsymbol{A}))^{-1}$.

When using the Hutchinson's trace estimator for the trace approximation we get the same result: $\mathbb{E}_{p(\boldsymbol{\epsilon})}[\int_0^1 \boldsymbol{\epsilon}^T \frac{\partial f}{\partial \boldsymbol{z}(t)} \boldsymbol{\epsilon} \, dt] = \mathbb{E}_{p(\boldsymbol{\epsilon})}[\boldsymbol{\epsilon}^T \boldsymbol{A} \boldsymbol{\epsilon}] = \text{tr}(\boldsymbol{A})$, where $\mathbb{E}(\boldsymbol{\epsilon}) = \boldsymbol{0}$ and $\text{Cov}(\boldsymbol{\epsilon}) = \boldsymbol{I}$.

Similarly, applying the discrete change of variables, we get the same result for the matrix exponential:

$$p(\boldsymbol{x}) = q(\boldsymbol{z})|\det J_F(\boldsymbol{z})|^{-1} = q(\boldsymbol{z})|\det \exp(\boldsymbol{A})|^{-1} = q(\boldsymbol{z}) \exp(\text{tr}(\boldsymbol{A}))^{-1}. \qquad (22)$$

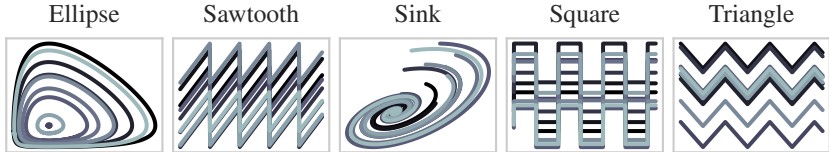

| Ellipse | Sawtooth | Sink | Square | Triangle |

Figure 5: Sample trajectories for synthetic data.

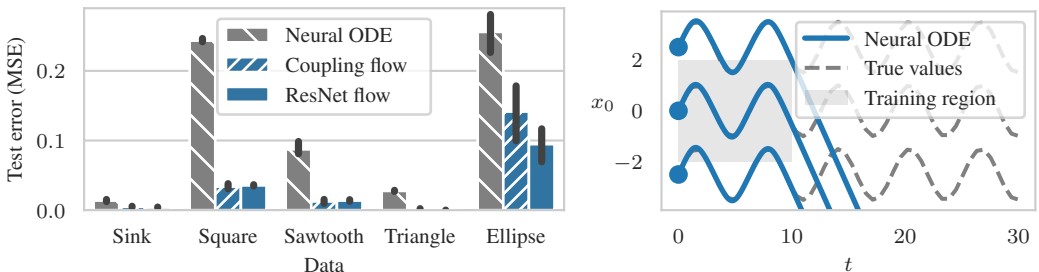

Figure 6: (Left) Test error for synthetic data. (Right) All models fail when extrapolating in time.

## A.8 Computation complexity of (continuous) normalizing flows

In general, evaluating the trace of the Jacobian of function $f : \mathbb{R}^d \to \mathbb{R}^d$ requires $O(d^2)$ operations. In CNFs, this operation has to be performed at every solver step. Since the number of steps can be very large for more complicated distributions [27], this becomes prohibitively expensive. Because of this, Grathwohl et al. [27] introduce computing the approximation of the trace during training. This has the benefit of having a lower cost, $O(d)$. The issue with this method is that the training becomes noisier and after training we have to again rely on exact trace to get the exact density.

On the other hand, computing the determinant of the Jacobian is $O(d^3)$ operation in general. Because of this, regular normalizing flows do not use unconstrained functions $f$, but rather opt for those that produce triangular Jacobians, e.g., autoregressive [41] or coupling transformations [17], where the determinant is just the product of the diagonal elements, i.e., it is of linear cost $O(d)$.

## B   Synthetic experiments

We first test the capabilities of our models on periodic signals:

- Sine: $f(t, x) = \cos(t)$ which corresponds to flow $F(t, x) = x + \sin(t)$, $x \in \mathbb{R}$,
- Sawtooth: $F(t, x) = x + t - \lfloor t \rfloor$,
- Square: $F(t, x) = x + \text{sign}(\sin(t))$,
- Triangle: $F(t, x) = \int_0^t \text{sign}(\sin(u)) \, \mathrm{d}u$.

We sample initial values $x$ uniformly in $(-2, 2)$ and set the time interval to $(0, 10)$. We additionally check how well the models extrapolate by extending the initial condition interval to $(-4, 4)$ and time to 30. We also use two datasets, generated as solutions to known ODEs:

- Sink: $f(t, \boldsymbol{x}) = \begin{bmatrix} -4 & 10 \\ -3 & 2 \end{bmatrix} \begin{bmatrix} x_1 \\ x_2 \end{bmatrix}$,

- Ellipse: $f(t, \boldsymbol{x}) = \begin{bmatrix} \frac{2}{3}x_1 - \frac{2}{3}x_1 x_2 \\ x_1 x_2 - x_2 \end{bmatrix}$, which is a particular parametrization of Lotka-Volterra equations, also known as predator-prey equations,

where we sample initial conditions $x_1, x_2 \in [0, 1]$ uniformly. For extrapolation, we use $x_1, x_2 \in [1, 2]$. Figure 5 shows the generated trajectories for all synthetic datasets.

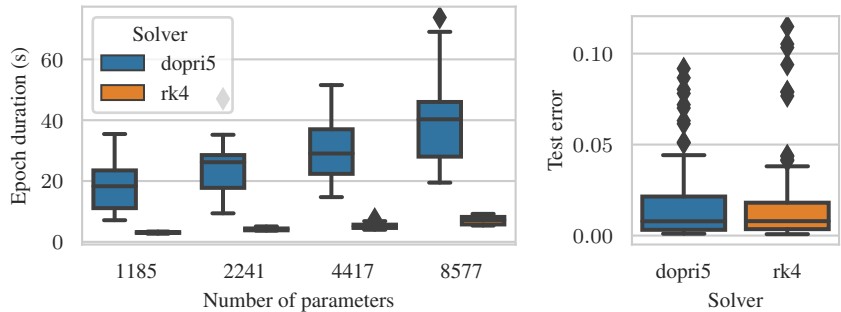

Figure 7: Fixed solvers are faster to train on synthetic data (Left) but they still have similar accuracy compared to adaptive solvers (Right).

## B.1 Comparing adaptive and fixed-step solvers

We ran an extensive hyperparameter search for Sine dataset. We test models with 2 or 3 hidden layers, each having dimension of 32 or 64, use $\tanh$ or ELU activations between them, and have $\tanh$ or identity as the final activation. For each of the model configurations we apply either no regularization or weigh the penalty term with $10^{-3}$. Finally, we run each trial 5 times with different seeds and compare between Runge-Kutta fixed-step solver with 20 steps and an adaptive 5th order Dormand-Prince method [18].

As expected, the vast majority of the trials fit the data very well. However, as Figure 7 shows, an adaptive solver always requires significantly longer training times, regardless of the size of the model, choice of the activations or regularization. We used default tolerance settings ($\text{rtol} = 10^{-7}$, $\text{atol} = 10^{-9}$) which is why we get such long training times. Therefore, in the other experiments, in the main text, whenever we use dopri5, we use $\text{rtol} = 10^{-3}$ and $\text{atol} = 10^{-4}$ to make training feasible. This once again shows the trade-off between speed and numerical accuracy.

From the results, one would expect that we can safely use fixed-step solvers and achieve similar or better results with smaller computational demand. However, as Ott et al. [63] showed, this can lead to overlapping trajectories which give non-unique solutions. Breaking the assumptions of our model can lead to misleading results in some cases. Here, we tackle density estimation with continuous normalizing flows as an example.

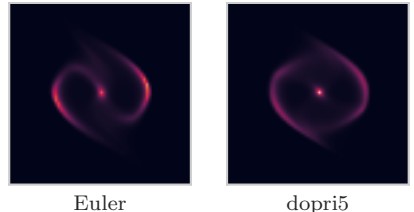

Euler          dopri5

Figure 8: Density learned with Euler and dopri5 solver. The estimated area under the curve for Euler method is 1.06, meaning it does not define a proper density.

We construct a synthetic 2-dim. dataset as a mixture of zero-centered normal distribution ($\sigma = 0.05$) and uniform points on the perimeter of a unit circle with small noise ($\sigma = 0.01$). We test adaptive dopri5 solver and Euler method with 20 steps.

The fixed solver achieves better results but Figure 8 visually demonstrates that it is not really capturing the true distribution better. It *cheats* by not defining a proper density function that integrates to 1. Since it has more mass to distribute, it can achieve better results. This might be hard to detect in higher dimensions and it can be particularly problematic since most of the literature reports log-likelihood on test data. Even though we took Euler method as an extreme example, the same can be shown for other solvers as well.

## B.2 Comparing flow configurations

Similar to Appendix B.1, we compare different flow models on synthetic sine data. We try coupling and ResNet models with linear and $\tanh$ for $\varphi$, as well as an embedding with 8 Fourier features (bounded to $(0, 1)$ interval in ResNet model), see Section 2.1 for more details. Both models have either 2 or 4 stacked transformations, each with a two hidden layer neural network with 64 hidden dimensions. We run each configuration 5 times with and without weight regularization ($10^{-3}$).

| MSE ($\times 10^{-2}$) | Ellipse | Sawtooth | Sink | Square | Triangle |
|---|---|---|---|---|---|
| Neural ODE | $25.59_{\pm 3.19}$ | $8.74_{\pm 1.10}$ | $1.38_{\pm 0.17}$ | $24.34_{\pm 0.3}$ | $2.76_{\pm 0.09}$ |
| Coupling flow | $14.16_{\pm 4.80}$ | $\mathbf{1.25}_{\pm \mathbf{0.33}}$ | $0.50_{\pm 0.06}$ | $\mathbf{3.38}_{\pm \mathbf{0.4}}$ | $0.19_{\pm 0.02}$ |
| ResNet flow | $\mathbf{9.48}_{\pm \mathbf{2.64}}$ | $1.38_{\pm 0.13}$ | $\mathbf{0.40}_{\pm \mathbf{0.04}}$ | $3.56_{\pm 0.1}$ | $\mathbf{0.0}_{\pm \mathbf{0.0}}$ |

Table 5: Test error on synthetic data, lower is better. Best results in bold.

| MSE ($\times 10^{-2}$) | Ellipse | Sawtooth | Sink | Square | Triangle |
|---|---|---|---|---|---|
| Neural ODE | $\mathbf{19.82}_{\pm \mathbf{1.34}}$ | $10.64_{\pm 1.76}$ | $18.0_{\pm 1.18}$ | $32.96_{\pm 3.0}$ | $4.22_{\pm 0.56}$ |
| Coupling flow | $515.8_{\pm 555.6}$ | $\mathbf{1.32}_{\pm \mathbf{0.36}}$ | $\mathbf{5.53}_{\pm \mathbf{2.23}}$ | $\mathbf{3.93}_{\pm \mathbf{0.76}}$ | $\mathbf{0.2}_{\pm \mathbf{0.04}}$ |
| ResNet flow | $100.4_{\pm 45.4}$ | $3.49_{\pm 1.14}$ | $6.65_{\pm 2.23}$ | $9.84_{\pm 2.94}$ | $0.79_{\pm 0.21}$ |

Table 6: Error on trajectories that start at initial conditions out of training distribution. Some trials returned outliers that skew the results (e.g., coupling flow on ellipse dataset).

| | Ellipse | Sawtooth | Sink | Square | Triangle |
|---|---|---|---|---|---|
| Neural ODE | $9.3_{\pm 0.88}$ | $8.25_{\pm 0.33}$ | $8.78_{\pm 0.81}$ | $7.81_{\pm 0.34}$ | $7.91_{\pm 0.35}$ |
| Coupling flow | $\mathbf{0.7}_{\pm \mathbf{0.11}}$ | $\mathbf{0.46}_{\pm \mathbf{0.22}}$ | $\mathbf{0.6}_{\pm \mathbf{0.05}}$ | $\mathbf{0.49}_{\pm \mathbf{0.14}}$ | $\mathbf{0.58}_{\pm \mathbf{0.16}}$ |
| ResNet flow | $1.05_{\pm 0.04}$ | $1.01_{\pm 0.15}$ | $1.24_{\pm 0.13}$ | $0.98_{\pm 0.04}$ | $1.01_{\pm 0.09}$ |

Table 7: Wall-clock time (in seconds) to run the last training epoch, using the same batch size.

All the models capture the data perfectly, except for the coupling flow with linear function of time $\varphi$ which does not converge. This could be due to inability of neural networks to process large input values. The issue can be fixed with different initialization or normalizing the input time values.

Tables 5, 6 and 7 show that neural flows outperform neural ODEs in forecasting, extrapolation with different initial values, and are faster during training.

## C  Additional results

Table 8 compares the training times for smoothing experiment. Neural ODE models use Euler method with 20 steps (the adaptive method is slower). Table 9 shows the average wall-clock time to run a single epoch for different TPP models. We include ablations for flow and ODE models that use different continuous RNN encoders, and a model without an encoder. Table 10 shows full negative log-likelihood results for the TPP experiment. Table 11 shows the full NLL results for marked TPPs.

| | Activity | MuJoCo | Physionet |
|---|---|---|---|
| Neural ODE | $200.884_{\pm 7.239}$ | $192.209_{\pm 2.526}$ | $103.198_{\pm 4.977}$ |
| Coupling flow | $\mathbf{106.298}_{\pm \mathbf{2.314}}$ | $\mathbf{46.171}_{\pm \mathbf{1.742}}$ | $\mathbf{78.561}_{\pm \mathbf{1.050}}$ |
| ResNet flow | $134.336_{\pm 3.453}$ | $102.745_{\pm 2.369}$ | $101.966_{\pm 8.285}$ |

Table 8: Average time (in seconds) to run a single epoch during training for different models, all other training parameters being the same.

| | | Poisson | Hawkes1 | Hawkes2 | Renewal | MOOC | Reddit | Wiki |
|---|---|---|---|---|---|---|---|---|
| Cont. | Neural ODE | 96.7 | 129.8 | 208.6 | 111.2 | 844.2 | 612.8 | 157.9 |
| | Coupling flow | 10.8 | 11.2 | 10.8 | 11.1 | 180.8 | 113.1 | 31.7 |
| | ResNet flow | 7.1 | 7.1 | 7.2 | 7.3 | 130.0 | 83.8 | 19.9 |
| Mixture | GRU-ODE | 39.7 | 42.3 | 55.9 | 39.3 | 600.0 | 419.5 | 97.9 |
| | ODE-LSTM | 35.9 | 39.0 | 37.8 | 43.8 | 569.4 | 443.6 | 109.4 |
| | Coupling flow | 3.4 | 3.4 | 3.3 | 3.3 | 47.0 | 37.2 | 8.5 |
| | ResNet flow | 5.9 | 5.9 | 5.8 | 5.9 | 96.5 | 64.9 | 16.1 |
| | GRU flow | 3.6 | 3.5 | 3.3 | 3.7 | 52.8 | 36.4 | 9.7 |

Table 9: Average time (in seconds) to run a single epoch during training for TPP models.

| Synthetic data | | Poisson | Hawkes1 | Hawkes2 | Renewal |
|---|---|---|---|---|---|
| | Ground truth | 0.9996 | 0.6405 | 0.1192 | 0.2667 |
| | Without history | 1.0046 | 0.7826 | 0.2354 | 0.2837 |
| | Discrete GRU | $1.0097_{\pm0.005}$ | $0.6424_{\pm0.006}$ | $0.1267_{\pm0.006}$ | $\mathbf{0.2598}_{\pm\mathbf{0.016}}$ |
| Cont. | Jump ODE | $\mathbf{0.9945}_{\pm\mathbf{0.016}}$ | $0.6461_{\pm0.009}$ | $0.2246_{\pm0.042}$ | $0.3124_{\pm0.022}$ |
| | Coupling flow | $1.0099_{\pm0.005}$ | $0.6441_{\pm0.007}$ | $0.1376_{\pm0.005}$ | $0.2720_{\pm0.017}$ |
| | ResNet flow | $1.0105_{\pm0.005}$ | $0.6426_{\pm0.007}$ | $0.1813_{\pm0.025}$ | $0.2851_{\pm0.018}$ |
| Mix. | GRU-ODE | $1.0100_{\pm0.005}$ | $\mathbf{0.6419}_{\pm\mathbf{0.007}}$ | $\mathbf{0.1239}_{\pm\mathbf{0.005}}$ | $0.2601_{\pm0.017}$ |
| | ODE-LSTM | $1.0108_{\pm0.005}$ | $0.6448_{\pm0.006}$ | $0.1253_{\pm0.005}$ | $0.2605_{\pm0.017}$ |
| | Coupling flow | $1.0103_{\pm0.005}$ | $0.6450_{\pm0.008}$ | $0.1254_{\pm0.006}$ | $0.2605_{\pm0.016}$ |
| | GRU flow | $1.0100_{\pm0.005}$ | $0.6439_{\pm0.007}$ | $0.1270_{\pm0.006}$ | $0.2608_{\pm0.016}$ |
| | ResNet flow | $1.0104_{\pm0.005}$ | $0.6443_{\pm0.006}$ | $0.1249_{\pm0.005}$ | $0.2603_{\pm0.017}$ |

| Real-word data | | MOOC | Reddit | Wiki |
|---|---|---|---|---|
| | Without history | 2.0623 | 1.5402 | 1.5813 |
| | Discrete GRU | $-0.4448_{\pm0.294}$ | $-0.9299_{\pm0.118}$ | $-0.5832_{\pm0.321}$ |
| Cont. | Jump ODE | $0.8710_{\pm0.157}$ | $0.1308_{\pm0.018}$ | $-0.3115_{\pm0.011}$ |
| | Coupling flow | $0.7694_{\pm0.172}$ | $-0.1263_{\pm0.273}$ | $-0.2807_{\pm0.500}$ |
| | ResNet flow | $\mathbf{-1.2379}_{\pm\mathbf{0.049}}$ | $\mathbf{-1.2962}_{\pm\mathbf{0.126}}$ | $-1.2907_{\pm0.045}$ |
| Mix. | GRU-ODE | $-0.2626_{\pm0.183}$ | $-1.0907_{\pm0.076}$ | $-1.3635_{\pm0.071}$ |
| | ODE-LSTM | $-0.2277_{\pm0.331}$ | $-1.0888_{\pm0.029}$ | $\mathbf{-1.3727}_{\pm\mathbf{0.327}}$ |
| | Coupling flow | $-0.4026_{\pm0.584}$ | $-1.0933_{\pm0.161}$ | $-1.2702_{\pm0.178}$ |
| | GRU flow | $-0.3509_{\pm0.220}$ | $-1.0605_{\pm0.113}$ | $-0.9852_{\pm0.105}$ |
| | ResNet flow | $-0.5664_{\pm0.278}$ | $-1.0291_{\pm0.174}$ | $-1.1937_{\pm0.048}$ |

Table 10: Test negative log-likelihood (mean±standard deviation) for all TPP models.

| | | MOOC | Reddit | Wiki |
|---|---|---|---|---|
| | Discrete GRU | $2.7563_{\pm0.141}$ | $1.8468_{\pm0.016}$ | $8.0527_{\pm0.170}$ |
| Cont. | Jump ODE | $4.6118_{\pm0.070}$ | $3.6654_{\pm0.000}$ | $10.6040_{\pm0.304}$ |
| | Coupling flow | $5.5494_{\pm0.413}$ | $3.6312_{\pm0.324}$ | $9.7214_{\pm0.101}$ |
| | ResNet flow | $2.9466_{\pm0.000}$ | $2.3932_{\pm0.131}$ | $10.4368_{\pm0.034}$ |
| Mix. | GRU-ODE | $3.5344_{\pm0.242}$ | $2.3078_{\pm0.033}$ | $\mathbf{7.5537}_{\pm\mathbf{0.065}}$ |
| | ODE-LSTM | $3.0723_{\pm0.114}$ | $1.9057_{\pm0.164}$ | $8.3187_{\pm0.231}$ |
| | Coupling flow | $\mathbf{2.5877}_{\pm\mathbf{0.176}}$ | $\mathbf{1.6817}_{\pm\mathbf{0.095}}$ | $8.8018_{\pm0.057}$ |
| | ResNet flow | $3.0005_{\pm0.081}$ | $1.9491_{\pm0.008}$ | $8.5489_{\pm0.267}$ |

Table 11: Test negative log-likelihood (mean±standard deviation) for all marked TPP models.

# D   Data pre-processing

## D.1   Encoder-decoder datasets

**MuJoCo dataset.** Using Deep Mind Control Suite and MuJoCo simulator, Rubanova et al. [69] generate 10000 sequences by sampling initial body position in $\mathbb{R}^2$ uniformly from $[0, 0.5]$, limbs from $[-2, 2]$, and velocities from $[-5, 5]$ interval. We use this dataset without any changes.

**Activity dataset.** Following [69], we round up the time measurements to 100ms intervals. This was done to reduce the size of the union of all the points when batching but is unnecessary when using our flow models, and also when using the reparameterization for ODEs [9].

Original labels are: walking, falling, lying down, lying, sitting down, sitting, standing up from lying, on all fours, sitting on the ground, standing up from sitting, standing up from sitting on the ground. Rubanova et al. [69] combine similar positions into one group resulting in 7 classes: walking, falling, lying, sitting, standing up, on all fours, sitting on the ground. Data is split in train, validation and test set (75%–5%–20%).

**Physionet dataset.** We use PhysioNet Challenge 2012 [73], where the goal is to predict the mortality of patients upon being admitted to ICU. We process the data following [69] to exclude time-invariant features, and round the time stamps to one minute. Each feature is normalized to $[0, 1]$ interval. Data is split the same way as for MuJoCo: 60%–20%–20%.

When reporting MSE scores for the reconstruction task we scale the result by $10^2$ for activity dataset and by $10^3$ for others, for better readability. This is equivalent to scaling the data beforehand.

## D.2 MIMIC-III and MIMIC-IV

We follow [16] for processing **MIMIC-III** dataset. We process MIMIC-IV in a similar vein.

The publicly available **MIMIC-IV** database provides clinical data of intensive care unit (ICU) patients at the tertiary academic medical center in Boston [36, 25]. It builds upon the MIMIC-III database and contains de-identified patient records from 2008 to 2019 [35]. We use version MIMIC-IV 1.0, which was released March 16th, 2021.

To preprocess the data, we first select the subset of patients who:

- are registered in the admissions table,

- stayed in the ICU for at least 2 days and no more than 30 days,

- are older than 15 years at the time of the admission, and

- have chart-event data available,

which leaves us with 17874 patients.

There are four types of data sources available for ICU patients: chart-events, inputs, outputs and prescriptions. The chart-events table contains the patient's routine vital signs as well as any additional information such as laboratory tests. The input table documents drugs administered to the patient through, e.g., solutions and the prescription table stores information about medication given in any other form. Lastly, the outputs table contains any output data from, e.g., a catheter for the patient during their ICU stay.

Because the medication in the input table is administered over time, the administered units and doses have to be unified and then split into entries which are spread out over time. We choose 30 minutes as our sampling window and, for all administered medications with duration longer than an hour, split them into fixed time injections.

For all other tables, we only keep the most commonly used entries:

- **Chart-events**: Alanine Aminotransferase, Albumin, Alkaline Phosphatase, Anion Gap, Asparate Aminotransferase, Base Excess, Basophils, Bicarbonate, Bilirubin, Calcium, Chloride, Creatinine, Eosinophils, Glucose, Hematocrit, Hemoglobin, Lactate, Lymphocytes, Magnesium, MCH, MCV, Monocytes, Neutrophils, pCO2, pH, pO2, Phosphate, Platelet Count, Potassium, PT, PTT, RDW, Red Blood Cells, Sodium, Specific Gravity, Total CO2, Urea Nitrogen and White Blood Cells.

- **Outputs**: Chest Tube, Emesis, Fecal Bag, Foley, Jackson Pratt, Nasogastric, OR EBL, OR Urine, Oral Gastric, Pre-Admission, Stool, Straight Cath, TF Residual, TF Residual Output and Void.

- **Prescriptions**: Acetaminophen, Aspirin, Bisacodyl, D5W, Docusate Sodium, Heparin, Humulin-R Insulin, Insulin, Magnesium Sulfate, Metoprolol Tartrate, Pantoprazole, Potassium Chloride and Sodium Chloride 0.9% Flush.

## D.3 TPP datasets

We follow previous works to generate and pre-process temporal point process data [61, 71, 44].

**Synthetic data.** We use 4 synthetic datasets, for each we generate 1000 sequences, each sequence containing 100 elements. We generate Poisson dataset with constant intensity $\lambda^*(t) = 1$; Renewal with stationary log-normal density function ($\mu = 1$, $\sigma = 6$); and two Hawkes datasets with the

conditional intensity $\lambda^*(t) = \mu + \sum_{t_i < t} \sum_j^M \alpha_j \beta_j \exp(-\beta_j(t - t_i))$, with $M = 1$, $\mu = 0.02$, $\alpha = 0.8$ and $\beta = 1$ (Hawkes1), or $M = 2$, $\mu = 0.2$, $\alpha = [0.4, 0.4]$ and $\beta = [1, 20]$ (Hawkes2).

**Reddit.** We use timestamps of posts from most active users to most active topic boards (subreddits) [44]. There are 984 unique subreddits that we use as marks. We have 1000 sequences in total, each sequence is truncated to contain at most 100 points. This is done to make training with ODE-based models feasible.

**MOOC** is a dataset containing timestamps of events performed by users in interaction with a learning platform [44]. There are 7047 sequences, with at most 200 events. We have 97 different mark types corresponding to different interaction types.

**Wiki** contains timestamps of edits of most edited pages from most active users [44]. There are 1000 pages (sequences) with at most 250 events, and 984 users that we use as marks.

In our implementation, we use inter-event times $\tau_i = t_i - t_{i-1}$ and for real-world data, we normalize them by dividing them with the empirical mean $\bar{\tau}$ from the training set $\tau_i \mapsto \tau_i/\bar{\tau}$. This can still yield quite large values so for better numerical stability during training, we use log-transform $\tau \mapsto \log(\tau + 1)$. We can think of log-transformation as a change of variables and include it in the negative log-likelihood loss using the probability change of variables formula (see Section 3.3).

### D.4 Spatial datasets

For spatial data used in time-dependent density estimation experiment, we used the datasets from Chen et al. [9] with the same pre-processing pipeline. See [9] for further details.

**Earthquakes** contains earthquakes gathered between 1990 and 2020 in Japan, with the magnitude of at least 2.5 [78]. Each sequence has length of 30 days, with the gap of 7 days between sequences. There are 950 training sequences, and 50 validation and test sequences.

**Covid** data uses daily cases from March to July 2020 in New Jersey state [77]. The data is gathered on county level and dequantized. Each sequence covers 7 days. There are 1450 sequences in the training set, 100 in validation and 100 in test set.

**Bikes** contains rental events from a bike sharing service in New York using data from April to August 2019. Each sequence corresponds to a single day, starting at 5am. The data is split in training, test and validation set: 2440, 300, 320 sequences, respectively.

All the spatial values are normalized to zero mean and unit variance. We also normalize the temporal component to $[0, 1]$ interval.

## E  Hyperparameters

All experiments: Adam optimizer, with weight decay 1e-4

**Smoothing experiments**

- Batch size: 100
- Learning rate: 1e-3 with decay 0.5 every 20 epochs
- Hidden layers: 3
- Models
  - ODE models
    - Solver: euler or dopri5
  - Flow models: ResNet or coupling flow
    - Flow layers: 1 or 2
    - $\varphi(t)$: tanh for ResNet and linear for coupling (used in all experiments)
- Datasets
  - MuJoCo
    - Encoder-decoder hidden dimension: 100-100
    - Latent state dimension: 20

- GRU dimension: 50
  - ○ Activity
    - Encoder-decoder hidden dimension: 30-100
    - Latent state dimension: 20
    - GRU dimension: 100
  - ○ Physionet
    - Encoder-decoder hidden dimension: 40-50
    - Latent state dimension: 20
    - GRU dimension: 50

**Filtering experiment**

- Batch size: 100
- Learning rate: 1e-3 with decay 0.33 every 20 epochs
- Hidden dimension: 64
- Datasets: MIMIC-III or MIMIC-IV
- ODE models
  - ○ Solver: euler or dopri5
  - ○ Hidden layers: 3
- Flow models: GRU flow or ResNet flow
  - ○ Flow layers: 1 or 4
  - ○ Hidden layers: 2

**TPP experiment** (With or without marks)

- Batch size: 50
- Learning rate: 1e-3
- Hidden dimension: 64
- Data: Reddit or MOOC or Wiki
- ODE models
  - ○ Models: continuous or mixture
    - Mixture models: ODE-LSTM or GRU-ODE
  - ○ Hidden layers: 3
- Flow models
  - ○ Models: continuous or mixture
    - Continuous models: ResNet or coupling flow
    - Mixture models: ResNet or coupling or GRU flow
  - ○ Flow layers: 1
  - ○ Hidden layers: 2
- RNN models: GRU

**Density estimation experiment**

- Batch size: 50
- Learning rate: 1e-3
- Hidden dimension: 64
- Models: time-varying or attentive (for both CNFs and NFs)
- Continuous normalizing flows
  - ○ Hidden layers: 4
- Coupling normalizing flows
  - ○ Base density layers: 4 or 8
  - ○ Time-dependent NF layers: 4 or 8