# OpenReview forum: "Neural Flows: Efficient Alternative to Neural ODEs"
_NeurIPS.cc/2021/Conference — NeurIPS 2021 Poster_

### Official Review · Reviewer_KFCA · 2021-07-13

**Rating:** 7
**Confidence:** 4

**Summary:**

The derivative function of an ODE defines a vector field over the input space (augmented with time). Neural ODEs learn such vector fields when solving initial value problems. The authors propose a method that directly learns the flow of such a vector field, by learning a function that respects the constraints of a proper flow. This method avoids the need to perform a costly integration operation like in NODEs while also performing similarly or better.

**Limitations And Societal Impact:**

The authors have addressed these in sufficient detail.

**Main Review:**

## Strong points

- The paper attacks a very important problem of Neural ODEs: they are often slow, especially if the time series is very long. The paper proposes a simple but clever solution to avoid the need to integrate for potentially very long times.
- The authors show how to derive a valid flow counterpart to the GRU-ODE model, which has been used extensively in the literature. This model could have a significant impact in practice for the applications of interest.
- The results presented in the paper are very extensive and look at a diverse set of tasks. The authors present convincing results that the method is indeed faster (as expected) and in certain settings performs better.
- The paper is clearly written and easy to follow. The models are described in sufficient detail and the authors do a good job reviewing the Neural ODE counterpart models they improve upon like CNFs, Latent NODEs etc.
- The section on limitations is nice and offers a fair description.

## Improvements and Questions
- The section on approximation capabilities is a bit hand-wavy and does not present any concrete theoretical results. I think the authors could be more upfront about this and include it as a direction for future work. The authors also mention analytic functions, however, this is a relatively constrained class of functions. It might be more natural to discuss $C^{\infty}$ functions or $C^{k}$ functions for some finite $k$. Note that not all $C^{\infty}$ functions are analytical. Regarding the limitations of Neural ODEs in learning any homomorphism (e.g. $x \mapsto -x $), these have also been studied in *On Second Order Behaviour in Augmented Neural ODEs* (https://papers.nips.cc/paper/2020/hash/418db2ea5d227a9ea8db8e5357ca2084-Abstract.html). Additionally, how do the authors see their method being extended to applications involving higher-order dynamics?
- Some parallels with Neural ODEs could definitely use a computational complexity analysis to ground the discussion and make it more precise. For instance, in Section 3.3 the authors claim that Equation (10) becomes "intractable" as the dimension increases. This is slightly exaggerated since the complexity of the trace operation is $\mathcal{O}(D^2)$, where $D$ is the dimension of the space. In contrast, based on Equation (11), the complexity of the proposed method seems to be  $\mathcal{O}(D)$ since the determinant involves only the diagonal elements, which is indeed better. I believe that stating these complexities directly would clear up any confusion that might arise from describing this in words.
- I am missing in the experimental evaluation an experiment involving stiff ODEs. To me, it seems an ideal setting for the proposed method since any ODE solver struggles with stiff ODEs and learning the solution directly should perform much better and also be significantly faster.

**Time Spent Reviewing:**

6

---

> ### Author Response · Authors · 2021-08-10
> **Response**
>
> We want to thank the reviewer for their feedback.
>
> *The section on approximation capabilities*
>
> We will remove the statement in line 120 in favour of a more rigorous treatment as suggested. In particular, we will explain the $L^p$-universal approximation property for flows and Neural ODEs [47, 70]. Even though ResNets can model any ODE since they essentially correspond to the Euler discretization, this may not be always relevant in practice. For example, we do not know how many layers we need to match the performance. Therefore, we focused on empirically showing that our method obtains better results while being faster.
>
> *Higher-order dynamics*
>
> We will include the reference to Norcliffe et al. (2020), especially in the context of discussion of augmented Neural ODEs. They consider a second order ODE which constrains the function in a certain way to define the acceleration along with velocity. Norcliffe et al. (2020) motivate their parameterization, similar to ANODE, to learn a broader family of mappings. As we mentioned in the paper (lines 129-132), we can achieve the same with our method through simple modifications, in particular by loosening the invertibility condition. Alternatively, we can do the dimension padding, same as ANODE. Due to limited space, we left the investigations of higher-order dynamics for future work and focused on more prominent real-world tasks of learning the unknown dynamic.
>
> *Including the trace computation complexity*
>
> The trace complexity is O(D^2) while our model calculates the diagonal in O(D). As you correctly pointed out, this is already a big advantage of our method. The competing baseline, continuous normalizing flow, needs to calculate the trace at each solver step. Since the number of solver steps in practice is in the order of hundreds and more, the overall method becomes intractable, in contrast to our method which scales linearly with data dimension. An additional drawback of CNFs is that the current implementations perform D sequential autograd operations, so the benefits of parallelism are reduced whereas we can fully utilize modern hardware. We will include this to make the discussion more precise and remove confusion.
>
> Note that in Figure 2 we compare the stochastic trace estimator with our method. Calculating the exact trace is even slower. Using competing CNFs becomes problematic when we require exact density, i.e. after training is finished. We do not encounter such issues with our method since we get the exact density right away.
>
> *Stiff differential equations*
>
> We agree that including the discussion on stiff differential equations further strengthens our argument. It is a known fact that (Neural) ODEs can fail when the equation is stiff. Since we do not use solvers at all, our method will not encounter such issues.
>
> We ran an experiment as in Ghosh et al. (2020) [Section 4.4] that confirms that Neural ODEs fail when learning the stiff ODEs. Using the same training setup, our model learns this function with ease, as expected. Again, our model is much faster since we get the solution in a single call, whereas the Neural ODEs invoke the solver. We will include these findings in the revised version.
>
> References:
> Norcliffe et al. (2020). On Second Order Behaviour in Augmented Neural ODEs. NeurIPS
> Ghosh et al. (2020). STEER: Simple temporal regularization for neural odes. NeurIPS

---

> > ### Comment · Reviewer_KFCA · 2021-08-16
> > **Response to Authors**
> >
> > Thank you for addressing my comments, in particular the one regarding adding a stiff ODE experiment!
> >
> > I am satisfied with the response from the authors and I am happy to recommend the paper for acceptance.

---

### Official Review · Reviewer_U5x8 · 2021-07-14

**Rating:** 8
**Confidence:** 4

**Summary:**

This work proposes directly modelling the solution trajectories of an ODE, circumventing the need for costly numerical ODE solvers. Specifically, they model the function which returns a solution to the initial value problem, and derive conditions this function must satisfy in order to implicitly induce a valid ODE.

Crucially, the authors explain that Neural ODE models are not very interpretable, unlike more traditional applications of ODEs. Since we aren’t able to use the dynamics for interpretability, it makes sense to not modely them explicitly, and instead model its integral (via IVP) directly. (This is more my interpretation than what the authors stated explicitly.)

The authors propose several architectures, including ResNet flow and GRU flow. The issue with these models is that these models can only be inverted approximately and require expensive iterative schemes. A third approach is to design architectures inspired by normalizing flows, which trades off some expressivity for greater efficiency. The authors also consider the approximation capabilities of these models.

Section 3 introduces specific models for time-series and density estimation, and Section 4 compares these models to baselines on the relevant tasks.


**Limitations And Societal Impact:**

The authors have adequately addressed the limitations and potential negative societal impact of their work.

**Main Review:**

Describe the strengths of the work:

The authors carefully consider the benefits and drawbacks of Neural ODEs, and propose an original method that greatly increases the speed of these models while not harmings its expressivity, and retaining many of the benefits of Neural ODEs, namely naturally handling continuous time and irregular time series.

Neural Flows are significantly faster than Neural ODEs and match performance across a number of tasks.

The authors demonstrate the superiority of their approach on a wide array of tasks and models.

Explain the limitations of this work:

It might be interesting to see image density estimation tasks (e.g. comparing to FFJORD). On the other hand, the chosen time-dependent density estimation task is one that Neural ODE/Flow solve much better than other competing approaches.

Correctness:
The authors incorporate recent techniques to improve the speed of Neural ODEs and ensure they have strong baselines when comparing to their method, both in terms of speed and performance.
They report results with confidence intervals over multiple runs, and tuned baseline models. They provide thorough details on all experiments.

Clarity:

Very excellent writing throughout the paper and the appendix.

Typos:
Line 686: parameterization

Relation to prior work:

This work thoroughly and clearly explains how it builds/is related to previous work.

Reproducibility:

The appendix has thorough details on data processing and hyperparameters for all models and datasets.

**Time Spent Reviewing:**

3.5

---

> ### Author Response · Authors · 2021-08-10
> **Response**
>
> We would like to thank the reviewer for their feedback.
>
> *Image density estimation tasks, e.g. comparison to FFJORD*
>
> As you correctly pointed out and as we discussed in the paper (lines 235-239), we are not comparing our flow models and continuous normalizing flows on traditional density estimation since, in that case, our model reduces to the discrete normalizing flow. There are existing results that show that specialized architectures such as GLOW (Kingma et al., 2018) outperform CNFs/FFJORD (Grathwohl et al., 2018) on image density estimation. On the other hand, our method proposes an alternative to CNFs on time-dependent density estimation so we decided to focus on this task instead.
>
> References:
> Kingma et al. (2018). GLOW: Generative flow with invertible 1x1 convolutions.
> Grathwohl et al. (2018). FFJORD: Free-form continuous dynamics for scalable reversible generative models.

---

> > ### Comment · Reviewer_U5x8 · 2021-08-25
> > **Reply**
> >
> > Thanks for your response. In light of the concerns raised by other reviews, I have decided to adjust my score to 8. In particular, while this work is very well-written and makes a clear contribution, the evidence provided is mainly experimental. A more detailed theoretical justification would be required to justify a score of 9. But overall I still think this is a fantastic paper, and a very important work for the Neural ODEs community.

---

### Official Review · Reviewer_ZAV1 · 2021-07-15

**Rating:** 7
**Confidence:** 5

**Summary:**

This paper presents a natural evolution of the Neural ODE framework where, instead of learning the function representing the derivative driving the evolution of the studied process, the authors directly parametrize the flow of the ODE.

**Limitations And Societal Impact:**

The limitations were clearly addressed in the original submission.

**Main Review:**

The main idea of the paper is quite straightforward and elegant and so are the proposed architectures for the flow. Its strengths lie in the simplicity of implementation of the proposed and in the fact that this framework can easily be expanded into other models and different parameterization.

It is worth noting, and this is mentioned in the paper, that some choices of the NODE solver can yield networks which are more or less identical to some of the flows described here (eg. an explicit Euler disrectization would yield the ResNet flow).

However there are still some aspects which are not convincing in my view:
- NODE is not really a well-defined model in the sense that it encompasses many instanciations which can behave quite differently depending for example on the choice of the numerical solver. Thus, it seems to me that for some processes which need a numerical scheme more involved than an explicit scheme to be accurately modelled, directly modelling the flow isn't as easy as is said in 2.2. Examples on stiff differential equations for example would be welcome to explore this aspect.
- Not unrelated to the point above, it would be interesting to include experiments where the learned equations model challenging physical phenomena. For example, reaction-diffusion phenomena or examples from fluid dynamics could allow a meaningful and informative comparison between different instanciations of NODE and the flow models.
- In this last context, the flow models should be also compared to the slightly different approach brought by the Neural Fourier Operator paper (Li et al., ICLR 2020). I find it surprising that this paper is not cited as it proposes an approach which is quite similar in its motivation and philosophy.
- The improvements brought by this new point of view are in some cases somewhat marginal in terms of accuracy and it would also be interesting to compare the stability / easiness of training neural flows with NODE models.

Overall, I think this paper has many merits and I wouldn't oppose it being accepted at the conference. However, I would advocate for it more strongly if some experiments were added, for example along what is suggested above.

Edit after rebuttal: I thank the authors for their comprehensive answer to my review. I am not fully convinced by the answer regarding Neural Fourier operators: While I agree that the two settings differ in generality, Neural Flows should be usable in the setting of (Li et al.) and this would give an interesting comparison to the reader. However, I think this paper brings enough new ideas and experiments to be accepted at the conference and will update my score accordingly.

**Time Spent Reviewing:**

3

---

> ### Author Response · Authors · 2021-08-10
> **Response**
>
> We thank the reviewer for their feedback.
>
> *Neural ODEs encompass many instantiations which behave differently (e.g. depending on the numerical solver)*
>
> To keep the scope of the paper manageable, we study Neural ODE models with the focus on real-world problems. This already includes many different settings as described in Section 3. Thus, the goal was to show that our method can be used as an alternative to Neural ODEs when we learn the unknown dynamic. Such a problem is tackled by the vast majority of the Neural ODE follow-up papers. In all of these cases, they used standard numerical solvers. Since we have demonstrated the benefits in these applications, we believe our contribution is already of great value to practitioners.
>
> *Section 2.2 and stiff differential equations*
>
> The statements and the related work in Section 2.2 are there to ensure that we can model ODE dynamics with flows. We will expand this section to further discuss the $L^p$-universal approximation property of flows and Neural ODEs. In the paper, we mainly focus on providing the empirical evidence that the flows can be used as an alternative to Neural ODEs.
>
> **It is true that (Neural) ODEs can fail when the equation is stiff. Since we do not use solvers at all, our method will not encounter such issues.** The only remaining question is whether we can learn such dynamics. We empirically show that we can. We ran an experiment as in Ghosh et al. (2020) [Section 4.4] where the true function is at first steep but then continues flat. We confirm that Neural ODEs fail to learn this stiff function, getting the MSE of >3. At the same time, our model learns this function with ease, as expected, with the MSE of around 0.2. Again, our model is much faster since we get the solution in a single call, whereas the Neural ODEs invoke the solver. We will include these findings in the revised version.
>
> *Modeling physical phenomena and comparison to Neural Fourier Operator (ICLR, 2020)*
>
> We will include the reference to the paper by Li et al. (2020). They propose an architecture where the solution to a PDE is predicted at fixed discrete time points, on a 2D grid. **Although Li et al. (2020) and our approach tackle similar problems, they cannot be compared directly.** The model by Li et al. (2020) is analogous to the classical discrete encoder-decoder architectures. Their contribution is the introduction of a neural Fourier operator which allows them to transform the data and scale to different resolutions. They essentially perform an image to image mapping. Our method, in contrast, learns the dynamic in continuous time. This is also reflected in the experiments where we use many different tasks involving irregularly-sampled time series. In future work, one can combine the method from Li et al. (2020) with our method to, e.g., output the solutions to a PDE in continuous time.
>
> Note that the time-dependent density estimation is similar to modeling physical phenomena, such as reaction-diffusion or fluid dynamics, since we have a smooth function on some space (in this case, a probability density) that continuously changes with time. It would be interesting to explore this further in future work.
>
> *Marginal improvements in terms of accuracy*
>
> The main goal of our paper is efficiency while retaining expressiveness. This is exactly what we have achieved. When designing the experiments we used networks of similar size for both the flows and ODEs, with the sole exception of the density estimation task (see discussion in Section 5). We achieved better performance in terms of accuracy and at the same time showed that our method is much faster. This demonstrates the practical relevance of our models. Since the gap in evaluation times is rather large, we could have bigger models and still be faster. In the paper, we opted out of doing this to focus solely on the speed difference given that our method outperforms ODEs.
>
> Nevertheless, to show that this is true we increased the number of layers in the ResNet flow model in the smoothing experiment (Activity dataset) where we currently have the best accuracy but it is within the margin of error. **The accuracy of a bigger model increases by 2-3%, meaning it clearly outperforms an ODE while still being faster.**
>
> *Stability and easiness of training*
>
> We noticed that our flow models are easy to train and often require less iterations to reach the same performance. In general, an important factor in stability is the initialization. Since all of the models are designed to be close to the identity mapping when initialized (using standard weight initialization schemes such as Xavier), we do not encounter numerical issues or high loss values. That is, the choice of the architecture, such as the one we have in a coupling flow, allows us not to worry about weight initialization. We will include the discussion and empirical results in the final version.
>
> Finally, we would like to point out that we have included the limitations as well as the discussion of the broader impact in Section 5.
>
> References:
> Li et al. (2020). Fourier Neural Operator for Parametric Partial Differential Equations. ICLR
> Ghosh et al. (2020). STEER: Simple temporal regularization for neural odes. NeurIPS

---

### Official Review · Reviewer_izsU · 2021-07-31

**Rating:** 6
**Confidence:** 4

**Summary:**

The paper proposes the idea of directly modelling the solution curves for an ODE using neural networks as opposed to using numerical ODE solver.

**Ethical Concerns:**

No concerns

**Limitations And Societal Impact:**

The authors adequately addressed the limitations and potential negative societal impact

**Main Review:**

**After rebuttal: Score updated to reflect that the reviewer recommends acceptance**


*Pros:*

- a reasonable idea, straightforward modification of the original Neural ODE; gives evidence of advantages of directly approximating the solution of the ODE

*Cons:*

[1] the novelty of the idea of modelling the solution using neural networks as opposed to the dynamics function f should be clarified (see previous work such as Smaoui &Al-Enezi, 2004; Kumar &Yadav, 2011 and Dua (2011));

[2] it is hard to understand why the paper on a new method needs such an extensive applications section, describing different well-known problem statements such as filtering or smoothing, it might be a good idea to keep focused to the original message that modelling the solution as opposed to the dynamics function is beneficial (see point [2] of the decision justification)

[3] Related to the previous points, the storytelling should also be improved with expanding on the state-of-the-art and contribution discussions (see point [1] in the decision justification below)
[4] multiple statements are not clear (see comments below).

The following points stand between the reviewer and the paper acceptance (*decision justification*):

[1] State-of-the-art description may not be sufficient according to the reviewer: it would be good to contrast the novelty in this paper against the state of the art as listed above; the related work description for this paper should tell about the previous approaches using neural network; the novelty against the state of the art needs to be emphasised

[2]The reviewer thinks that the authors should stay focused in the main text on the paper idea, i.e. improving the efficiency of neural ODE and addressing the limitations of using numerical solvers; this work goes into extensive details on filtering and smoothing problem statements (section 3.1) which may look unnecessary long; the details of the encoder-decoder (section 3.1), which look similar to Neural ODE architecture, could be omitted with the reference made to the original Neural ODE paper, or moved to the appendix.

[3]Further comments that are given below on clarity and reproducibility (esp. comments [3],[5], [6], [7] below).

[4] It is not clear when in the experimental section (section 4) the Euler solver is used, what is its step size, and what is the reason behind comparing with the Euler solver when the original Neural ODE paper uses Dormand-Prince solver (dopri5); the provided code could partially help with it but it does not answer the question how the actual experiments in the paper have been done.

*Comments:*

[1]Figure 1: while it is understandable what the authors are saying in this image, not sure that this figure is actually conveying the idea that the approach eliminates the need for the numerical solvers. E.g. it would be good to state exactly what is the meaning of the arrows on the left-hand size image (I guess the iterations of the numerical solver?). Comparing it with the original Neural ODE paper (Chen et al, 2018, Figure 1), I can see that Chen et al use the arrows to denote the vector field direction (which is apparently different from the meaning of arrows in Figure 1 of this paper), and they state it explicitly. They also state that ‘circles represent evaluation locations’; similar clarifications should be made for Figure 1.

[2](As it is the appendix, it did not influence the score)  As a suggestion, formatting in section E (Hyperparameters) could be improved by placing these hyperparameters in a table.

[3] Ln 119: "Historically, the Cauchy-Kovalevskaya theorem [13, p. 39] tells us that, if the derivative f is an analytic function, then the solution curve F is analytic too. This hints that as long as we have expressive enough networks, we can approximate any ODE.” Not sure the reviewer sees the proposed connection between expressivity of the networks and the dynamics function being analytic. This is because the Universal approximation theorem (Cybenko, 1989) is stated for continuous functions and not analytic, and therefore the connection to the Cauchy-Kovalevskaya theorem is not clear to the reviewer. Therefore, the justfication could be stronger.

[4]Ln 103: "The disadvantage of both ResNet flow and GRU flow is the missing analytical inverse.” While it is clear that the invertibility would ensure the uniqueness of the solution, it is unclear why the analytical inverse is necessary for the proposed method.

[5]   Ln. 234: "Again, if we consider a linear ODE, we can easily show that calculating the trace and calculating the determinant of the corresponding flow is equivalent.” How exactly could it be shown? Is it possible to for example, give a proof in the appendix?

[6] Ln 79: "Although plain ResNets are not invertible, one could use spectral normalization [25]  to enforce a small Lipschitz constant of the network, which guarantees invertibility [65, Theorem 9.23]” Theorem 9.23 in [65] discusses the existence of the fixed point in the contraction of X into X, and it is not clear from this description why it would imply invertibility. The reviewer recommends referring to the justification in [2], which actually describes the construction of Lipschitz constants.

[7] Figure 2: it is unclear what experiment this figure is based upon (as the authors use three datasets in this section, MuJoCo, Physionet and Activity). Furthermore, as I can see in the description for Figure 2, Neural ODE is run with an Euler method, while the original Neural ODE paper reports results with dopri5 solver, why was it done this way?

Smaoui, N., & Al-Enezi, S. (2004). Modelling the dynamics of nonlinear partial differential equations using neural networks. Journal of Computational and Applied Mathematics, 170(1), 27-58.

Kumar, M. and Yadav, N., 2011. Multilayer perceptrons and radial basis function neural network methods for the solution of differential equations: a survey. Computers & Mathematics with Applications, 62(10), pp.3796-3811.

Dua, V. (2011). An artificial neural network approximation based decomposition approach for parameter estimation of system of ordinary differential equations. Computers & chemical engineering, 35(3), 545-553.

Cybenko, G. (1989). "Approximation by superpositions of a sigmoidal function". Mathematics of Control, Signals, and Systems.

**Time Spent Reviewing:**

6

---

> ### Author Response · Authors · 2021-08-10
> **Response**
>
> We thank the reviewer for their feedback. In the following we address the raised issues.
>
> *1) Discussion of additional related work: Smaoui & Al-Enezi (2004); Kumar & Yadav (2011); Dua (2011)*
>
> We thank the reviewer for the pointers to these articles and will further expand the discussion to include the additional references. We would like to point out that the main goal of our paper is to propose a computationally efficient alternative to Neural ODEs. As such, the baselines we use need to satisfy certain properties, namely the uniqueness of solution given an initial condition. Thus, **not all related work is directly comparable**. Additionally, as the original Neural ODE paper and its follow-up work suggest, modeling the dynamics of unknown underlying ODEs is the main use case, in contrast to approximating the solutions to a specific given ODE.
>
> Smaoui & Al-Enezi (2004) model two specific equations, Kuramoto-Sivashinsky and Navier-Stokes, by using dimensionality reduction and predicting the discrete future time steps. On the other hand, our method models arbitrary ODEs and works on a continuous time domain.
>
> Kumar & Yadav (2011) review existing approaches to approximating the ODE solutions. In our paper, we cite some of the papers mentioned by Kumar & Yadav (2011), e.g. Lagaris et al. [44] whose method we generalize and Jianyu et al. [48] that approximate linear ODEs with radial basis functions, in contrast to our method where we learn the unknown dynamics. The works in Kumar & Yadav (2011) use low-dimensional equations and focus on approximating the solutions which is not the primary goal of Neural ODEs and consequently is not the main goal of our work either. Another difference is that we require valid flows, exactly corresponding to some ODEs, in contrast to other related works. Therefore, we compare to the state of the art Neural ODE papers using large datasets and learn high-dimensional nonlinear ODEs.
>
> Similar to [61] which we cite in our paper, Dua (2011) model the solutions directly with neural networks. Since this again does not match the needed properties, which are required in all of our experiments, we could not include a comparison.
>
> *2) Extensive application section*
>
> In our thorough study we include many experiments so the goal of Section 3 is to provide a self-contained and fully reproducible setup as well as to clearly indicate how we replace ODEs with our models. We believe this is useful for readers who are not experts in the field. Some parts can be further shortened and we will try to keep the focus on the main message in the revised version.
>
> *4) Using Euler solver*
>
> We would like to point out a misunderstanding, **we do not only use an Euler solver**, in fact we always use dopri5 and we additionally compare it to Euler in some experiments. This is advantageous for Neural ODEs since the Euler method is faster than dopri5. However, even with this clear advantage, our method is still faster than both, as we have shown. Note that we also use a reparameterization trick [7] and a semi-norm trick [37] to further speed-up the ODEs, thereby improving the original implementations. We discuss and compare the solvers in the main text and Appendix (lines 269-271, 287, Appendix B.1 and E).
>
> *3) Other comments*
>
> Line 103: Although in many cases we do not need an analytical inverse, we need it when the initial time is not 0 (which we discuss in lines 61-66) and when using density estimation, as is discussed and demonstrated in Section 3.3 and experiments.
>
> Line 119: The Cauchy-Kovalevskaya theorem serves as an illustrative example. We will remove this statement in favor of a more rigorous treatment that was laid out by the existing works (lines 121-125). In particular, we will explain the $L^p$-universal approximation property for flows and Neural ODEs [47, 70].
>
> Line 234: Since the flow of a linear ODE is a matrix exponential, and the determinant of a matrix exponential is the exponential of the matrix trace, both versions (continuous and discrete) of the change of variables formulas (Section 3.3) are equivalent.
>
> Line 79: We will use the suggested reference.
>
> Figure 1: The left figure represents the vector field (colored arrows), solution points (black dots), derivative in a single point (black arrow), and solution curve (black curve). The right figure shows solution curves (colored curves), initial condition and solution point (black points) and solution curve (black curve). We did not show the solver points on the left figure but this would indeed demonstrate the difference between having a solver and our approach. We will clarify this in the paper.
>
> Figure 2: We show all the datasets for all the experiments, i.e. MuJoCo, Activity, Physionet, Mimic-III, Mimic-IV, MOOC, Reddit, Wiki, Bikes, Covid and Earthquakes. Due to the limited space we omitted the labels for datasets but we will fix this in the final version. As we already pointed out, we did not solely report results with the Euler method. Every experiment used dopri5 and in addition to this we included Euler to compare the performance. Sometimes Euler gave better results in terms of accuracy. As we discussed previously, Euler also gives better runtime results compared to adaptive methods but we still outperform it with our method. If we only reported dopri5 results, which would be a completely fair comparison, the gap would be even bigger.
>
> We will improve the formatting in Appendix E.
>
> References:
> Smaoui & Al-Enezi (2004). Modelling the dynamics of nonlinear partial differential equations using neural networks.
> Kumar & Yadav (2011). Multilayer perceptrons and radial basis function neural network methods for the solution of differential equations: a survey.
> Dua (2011). An artificial neural network approximation based decomposition approach for parameter estimation of system of ordinary differential equations.

---

> > ### Comment · Reviewer_izsU · 2021-08-17
> > **Re: questions sufficiently addressed; recommending acceptance**
> >
> > First of all, I would like to thank the authors for elaborate responses on the questions of myself and other reviewers.
> >
> > I am happy to say that the decision justification questions are resolved by this response the following way, which allows me to update my scores and recommend acceptance:
> >
> > [1] The discussion in the response clearly addresses my concerns, contrasting the previous attempts to directly model the ODE curves to the proposed method and discusses the limitations of the state-of-the-art methods.
> >
> > [2] As the authors responded to the concerns and agreed to emphasise their main message, I am happy with the improvement proposed by the authors.
> >
> > [3] The questions on clarity are also addressed.
> >
> > [4] I agree that the questions about the Euler solver are sufficiently addressed by the authors.

---

### Decision · Program_Chairs · 2021-09-27

**Decision:**

Accept (Poster)

**Comment:**

The paper proposes to directly model the solution of ODEs using time-conditional discrete flows, instead of via ODE integrators.
This has the potential to substantially accelerate applications of neural ODE flows. While the proposed model is rather straightforward, the extensive experiments highlight the nice properties of the approach.